# Multiomics and cellular senescence profiling of aging human skeletal muscle uncovers Maraviroc as a senotherapeutic approach for sarcopenia

Yang Li [1,2,7], Chuhan Li[1,7], Qin Zhou[1], Xingyuan Liu[3], Yulong Qiao[3], Ting Xie[4], Hao Sun [3,5,8], Michael Tim-Yun Ong [2,6,8] & Huating Wang [1,2,8] ✉

Cellular senescence is a hallmark of organismal aging but how it drives aging in human tissues is not fully understood. Here we leverage single nucleus multiomics to profile senescence in mononucleated cells of human skeletal muscle and provide the first senescence atlas. We demonstrate the intra- and inter-populational transcriptomic and epigenomic heterogeneity and dynamics of cellular senescence. We also identify commonalities and variations in senescence-associated secretory phenotypes (SASPs) among the cells and elucidate SASP mediated cellular interactions and niche deregulation. Furthermore, we identify targetable SASPs and demonstrate the possibility of using Maraviroc as a pharmacological senotherapeutic for treating age-associated sarcopenia. Lastly, we define transcription factors that govern senescence state and SASP induction in aging muscle and elucidate the key function and mechanism of JUNB in SASP activation. Altogether, our findings demonstrate the prevalence and function of cellular senescence in skeletal muscle and identify a novel pharmacological intervention for sarcopenia.

The organismal aging process is both characterized and driven by cellular and molecular mechanisms termed hallmarks of aging. Cellular senescence, an adaptive response induced by multiple physiological and pathological stresses that entails irreversible cell cycle arrest and resistance to apoptosis is emerging as a dominant mechanism orchestrating various facets of aging physiology and development of many age-dependent diseases[1–3]. Senescent cells (SnCs) are associated with multiple phenotypic and molecular changes that help distinguish them from quiescent or terminally differentiated cells, for example increased lysosomal β-galactosidase activity (SA-β-GAL), upregulated expression of cell cycle inhibitors, P21$^{CIP1/WAF1}$ (CDKN1A) and P16$^{INK4a}$ (CDKN2A)[4]. More importantly, SnCs are characterized by the senescence-associated secretory phenotype (SASP), which encompasses a variety of secreted bioactive factors that can include pro-inflammatory cytokines, chemokines, growth factors and matrix-remodling enzymes[2,5]. These SASPs are the major mediators of the paracrine effects of SnCs in their tissue

[1]Department of Orthopaedics and Traumatology, Li Ka Shing Institute of Health Sciences, The Chinese University of Hong Kong, Hong Kong SAR, China. [2]InnoHK Center for Neuromusculoskeletal Restorative Medicine, Hong Kong Science Park, Hong Kong SAR, China. [3]Department of Chemical Pathology, Li Ka Shing Institute of Health Sciences, The Chinese University of Hong Kong, Hong Kong SAR, China. [4]Center for Tissue Regeneration and Engineering, Division of Life Science, Hong Kong University of Science and Technology, Hong Kong SAR, China. [5]Warshel Institute for Computational Biology, Faculty of Medicine, The Chinese University of Hong Kong, Shenzhen, Guangdong, China. [6]Department of Orthopaedics and Traumatology, The Chinese University of Hong Kong, Hong Kong SAR, China. [7]These authors contributed equally: Yang Li, Chuhan Li. [8]These authors jointly supervised this work: Hao Sun, Michael Tim-Yun Ong, Huating Wang. ✉e-mail: huating.wang@cuhk.edu.hk

microenvironment and can disrupt tissue homeostasis by promoting chronic inflammation, fibrosis and progenitor cell dysfunction. For example, SASP factors contribute to the chronic inflammation in aging tissue (so-called inflammaging) by directly initiating inflammation or indirectly inducing secondary inflammation through chronic activation of immune cells and spread of senescence locally and systemically.

It is becoming clear that the heterogeneity of SnCs and SASPs is vast, yet ill-characterized[5,6]; for example, the extent of the expression of different SASP elements is highly cell-type and senescence-stage dependent. Even less is known about the regulation of senescence state and SASPs particularly the upstream transcriptional regulators/events that activate the SASPs. NF-κB signaling serves as a pivotal regulator of SASP interleukins and cytokines, chemokines, growth factors and other factors[6–8]; CCAAT/enhancer-binding protein-beta (C/EBPβ) functions as a TF that cooperates with NF-κB to modulate the expression of some SASPs[9–11], but there remains an imperative need to define additional orchestrators and elucidate their regulatory mechanisms in governing SASP expression. Emerging single-cell technologies have enormously facilitated the profiling and characterization of senescence/SASPs heterogeneity, dynamics and regulation spatiotemporally. However, most studies so far are from model animals and there is sparse information about the prevalence and spectrum of SnCs/SASPs in human tissues such as skeletal muscle.

Skeletal muscle, as a key organ of body homeostasis and mobility, suffers from age-associated sarcopenia, which involves a deterioration in muscle quantity and quality, muscle strength and muscle function[12]. It is believed that the condition can partly be attributed to the decreased number and function of adult muscle stem cells (MuSCs)[13,14]. MuSCs are indispensable for maintaining muscle homeostasis and injury-induced muscle regeneration. In aged mice, muscle repair is blunted in a large part due to the numeric and functional decline of MuSCs. Accumulated works demonstrate both intrinsic alterations in MuSCs and extrinsic deregulations in the niche microenvironment contribute to the age-related decline. The intricate interplay between the niche and MuSCs is emerging as a key question in understanding molecular mechanisms underlying muscle aging. Moreover, burgeoning evidence suggests signs of senescence in skeletal muscle but conflicting observations are being reported[12,15]. For example, earlier studies reported signs of senescence in isolated MuSCs from aged mice (elevated expression of SA-β-GAL and p16, p21, and Igfbp5 mRNAs)[16], but more recent studies revealed a lack of conclusive evidence for senescence-associated phenotypes such as SA-β-GAL staining on aged muscle sections[17,18]. Nevertheless, several studies showed that genetic elimination of SnCs restored muscle loss and inflammation in aged mice[19,20]. Therefore, the prevalence of the senescence in MuSCs and also in other resident mononuclear cells that populate the interstitial microenvironment of skeletal muscle remains elusive, let alone the possible contribution of senescent MuSCs in niche deregulation of aging muscle.

It is now well believed that SnCs are therapeutically targetable; a relatively new class of drugs termed senotherapeutic approaches that selectively kill senescent cells (senolytics) or to suppress SASPs (senomorphics) are attracting unprecedented attention as a means to enable healthy aging[6,21]. Indeed, in preclinical models, pharmacological elimination of senescent cells restores health and youthful properties in multiple tissues. Of note, senolytic treatment with dasatinib/quercetin also increased muscle strength and function in aged mice[22]. Other studies, however, suggested that pro-senescence therapy can promote muscle regeneration[23]. Overall, we must admit that our knowledge of senescence in aging muscle is largely incomplete; we believe it is imperative to start from the ab initio defining of the unique features of SnC atlas in skeletal muscle and provide a full spectrum of senescence/SASP heterogeneity, dynamics, and regulation, which will enable the exciting opportunity for identification of novel senotherapeutic targets and drug discovery/development for sarcopenia.

Here in this study, we conducted simultaneous single-nucleus RNA-seq + ATAC-seq mapping in mononucleated cells freshly harvested from young and aged human biopsies. Integrated analysis yields the first senescent cell atlas in aging human muscle and demonstrates the SASP heterogeneity and dynamics. We further elucidate the function of senescent cells, in particular the contribution of senescent MuSCs in niche modulation via SASP-mediated cell-cell interactions. Moreover, we test the possibility of senotherapeutic targeting in aging muscle and identify Maraviroc as an effective senomorphic approach for ameliorating muscle aging in mice. Furthermore, we elucidate upstream TFs governing senescence and define JUNB as a direct transcriptional activator of SASP induction via enhancer regulation in senescent MuSCs. The finding demonstrates the key role of JUNB in SASP regulation and highlights it as a potential senotherapeutic target.

## Results
### Multiomics mapping of senescence atlas in human muscle
To gain the first senescence blueprint of mononucleated cells in aging human muscle, we obtained hamstring muscles from biopsies of 10 male donors: five young (19–27 years-old males who underwent anterior cruciate ligament reconstruction) and five aged (60–77 years-old males who underwent knee replacement surgery) (Fig. 1 and Suppl. Dataset 1). Structurally intact and histologically healthy hamstring muscles were harvested from anatomically equivalent regions across cohorts and processed under standardized protocols. Mononuclear cells were FACS isolated and single nuclei were prepared for single-nucleus (sn) multiomics (simultaneous RNA-seq and ATAC-seq in one cell) analysis using a 10x Genomics Chromium. After stringent quality control to remove low-quality nuclei and batch effect, a total of 52,934 (30,390 from five young and 22,544 from five aged donors) qualified nuclei were obtained for downstream analysis (Suppl. Fig. 1A). The snRNA-seq data quality, indicated by the number of read counts (UMI) and genes with at least one read count was highly correlated across all five donors (Suppl. Fig. 1B); similar observation was made on the ATAC-seq data quality as indicated by ATAC read counts and ATAC peaks for each sample (Suppl. Fig. 1C). Using unbiased clustering and uniform manifold approximation and projection (UMAP) analysis, 12 clusters of muscle resident mononuclear cells were identified with distinct transcriptomic and epigenomic signatures, including MuSCs, FAP (Fibro-adipogenic progenitors) 1, FAP 2, EC (endothelial cells) 1, EC 2, EC 3, Pericytes, SMC (smooth muscle cells) 1, SMC 2, MPs (macrophages), and B/T/NK (B-cells, T-cells and Natural Killer cells), along with a small number of nuclei from mature skeletal muscles (MSMs) despite terminally differentiated and post-mitotic myofibers were excluded from the procedure (Fig. 2A, Suppl. Fig. 1D and Suppl. Dataset 2). The marker genes from the literature were used to annotate these clusters, for example, MYF5 and PAX7 for the MuSCs, DCN, FBN1 for the FAPs, ACTA1 for the ECs, and C1QA for the MPs (Suppl. Fig. 1E-F and Suppl. Dataset 2). Strong correlations in global gene expression were observed across all 10 donors within each cell type (Suppl. Fig. 1G), suggesting that the within-group samples were highly comparable. Consistently, all the expected cell types were identified among donors in each age group with comparable relative abundance (Suppl. Fig. 1H-I). Decreased numbers of MuSCs, ECs, SMCs, and Pericytes were observed while the numbers of MSCs, FAPs, MPs, and B/T/NK cells increased in the aged compared to the young muscles (Fig. 2B), confirming the altered cellular composition and niche microenvironment during muscle aging. For the four major non-immune cell types, MuSCs, ECs, FAPs and SMCs, correlation analysis demonstrated that the cell proportions of MuSCs, ECs, and SMCs decreased, and FAPs increased with aging (Fig. 2C), which was consistent with previous findings[24,25]. The cell type prioritization score analysis using Augur pinpointed B/T/NK cells (0.90), SMCs (0.86), MuSCs (0.84), and MSMs (0.82) as the cells most responsive to aging (Fig. 2D). Furthermore, by

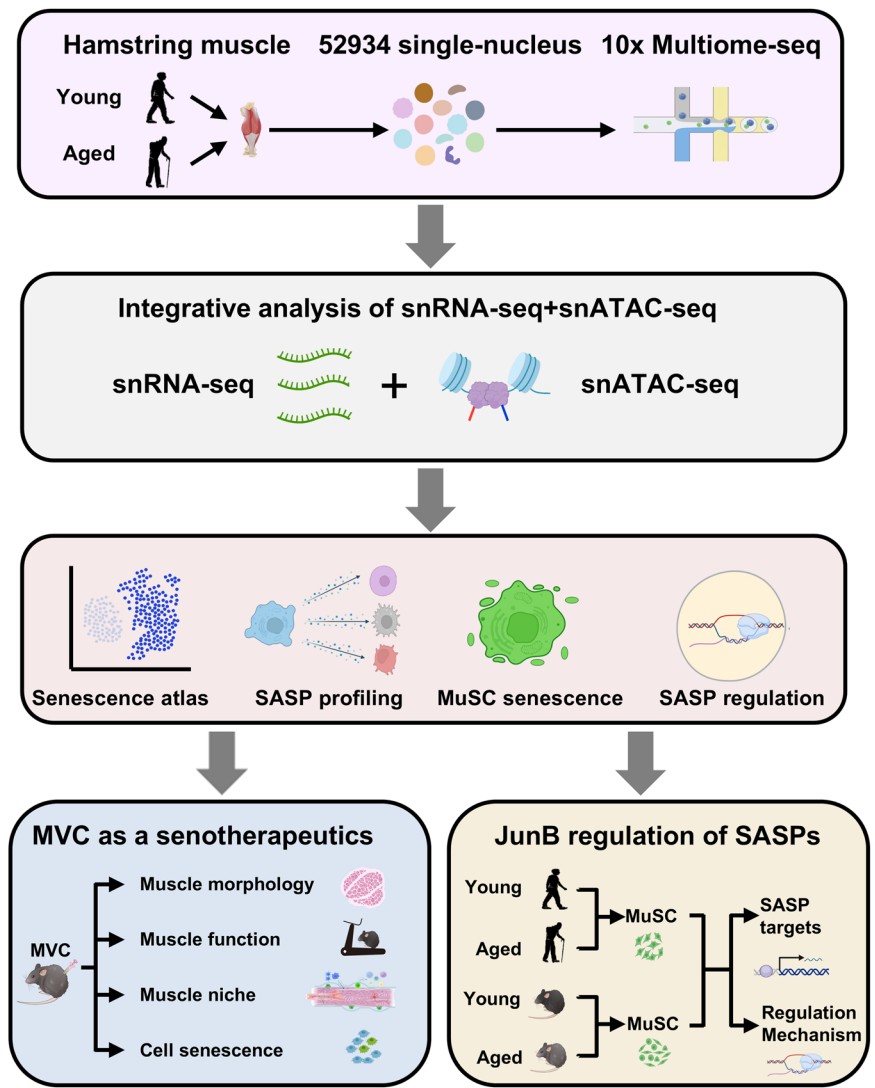

**Fig. 1 | Schematic of the overall design of the study.** Created in BioRender. Li, Y. (2025) https://BioRender.com/xsxka6w.

quantifying transcriptional noise, we found most cells showed an elevated transcriptional heterogeneity among individual nuclei (Fig. 2E), which is in line with the known feature of aging[26]. This finding was further validated by calculating the average transcriptional noise scores per donor and indeed an elevated ratio of transcriptional noise was observed in the aged group (Suppl. Fig. 1J).

Next, to define the senescence atlas, we calculated a unified senescence score (USS) by integrating four senescent gene sets (SenMayo, CellAge, GenAge, and Senescence Eigengene, Suppl. Dataset 2) through single-sample Gene Set Enrichment Analysis (GSEA)[27–31]. As a result, increased percentages of senescent cells were detected in all four cell types in aged vs. young muscle: MuSCs (12.2% vs. 7.9%, Fig. 2F), FAPs (26.9% vs. 3.3%, Fig. 2G), ECs (13.3% vs. 9.9%, Fig. 2H) and SMCs (40.3% vs. 4.7%, Fig. 2I). Gene Ontology (GO) enrichment analysis revealed that compared to the non-senescent (nSn) cells, the senescent (Sn) cells were enriched for pathways such as "wound healing", "response to oxidative stress", "cell adhesion" and "cell migration" etc. (Fig. 2J and Suppl. Dataset 3), all of which are characteristic features of senescent cells[32]. The above findings from analyzing the snRNA-seq data were validated by additional experiments (Fig. 2K). H&E staining of the muscle sections from additional pairs of donors revealed decreased fiber size and increased inflammation in aged vs. young muscles (Fig. 2L); elevated staining of P16 and P21 proteins was also detected (Fig. 2M). Moreover, total MuSCs

were freshly isolated by FACS and a significant increase of SA-β-GAL+ cells (Fig. 2N) and P16+ cells (Fig. 2O) were detected in aged muscles, accompanied by higher mRNA levels of senescence markers, *P14, P16, P19* and *P21* (Fig. 2P), altogether supporting the increased senescence in aged MuSCs.

## Heterogeneity and dynamics of cellular senescence in human muscle

To further characterize MuSC senescence, pseudotime trajectory was utilized to reveal the MuSC fate which diverged into two late paths (Fig. 3A). The Early-Late 1 fate accurately captured the transition from young to aged nuclei, showing a significant increase in the proportion of aged nuclei along the pseudotime (Fig. 3B and Suppl. Fig. 2A); while in the Early-Late 2 path, young and aged nuclei randomly distributed (Fig. 3B and Suppl. Fig. 2B). Nevertheless, senescent MuSCs accumulated in both late branches (Fig. 3C) with *CDKN1A* gene highly expressed at the ends (Fig. 3D). We then aggregated the expression of published cell cycle gene set of Reactome collection (Suppl. Dataset 3) along the pseudotime trajectory and calculated a module score by ss-GSVA method[33]; a decreased ss-GSVA score was observed over the trajectory (Fig. 3E), in agreement with the known feature of cell cycle arrest in senescent cells[34,35]. Differentially expressed gene (DEG) analysis uncovered various senescence-related GO terms such as "Extracellular space" (such as *FGF2, IGFBP6, IGFBP7, CXCL12, TGFB1, BMP6*),

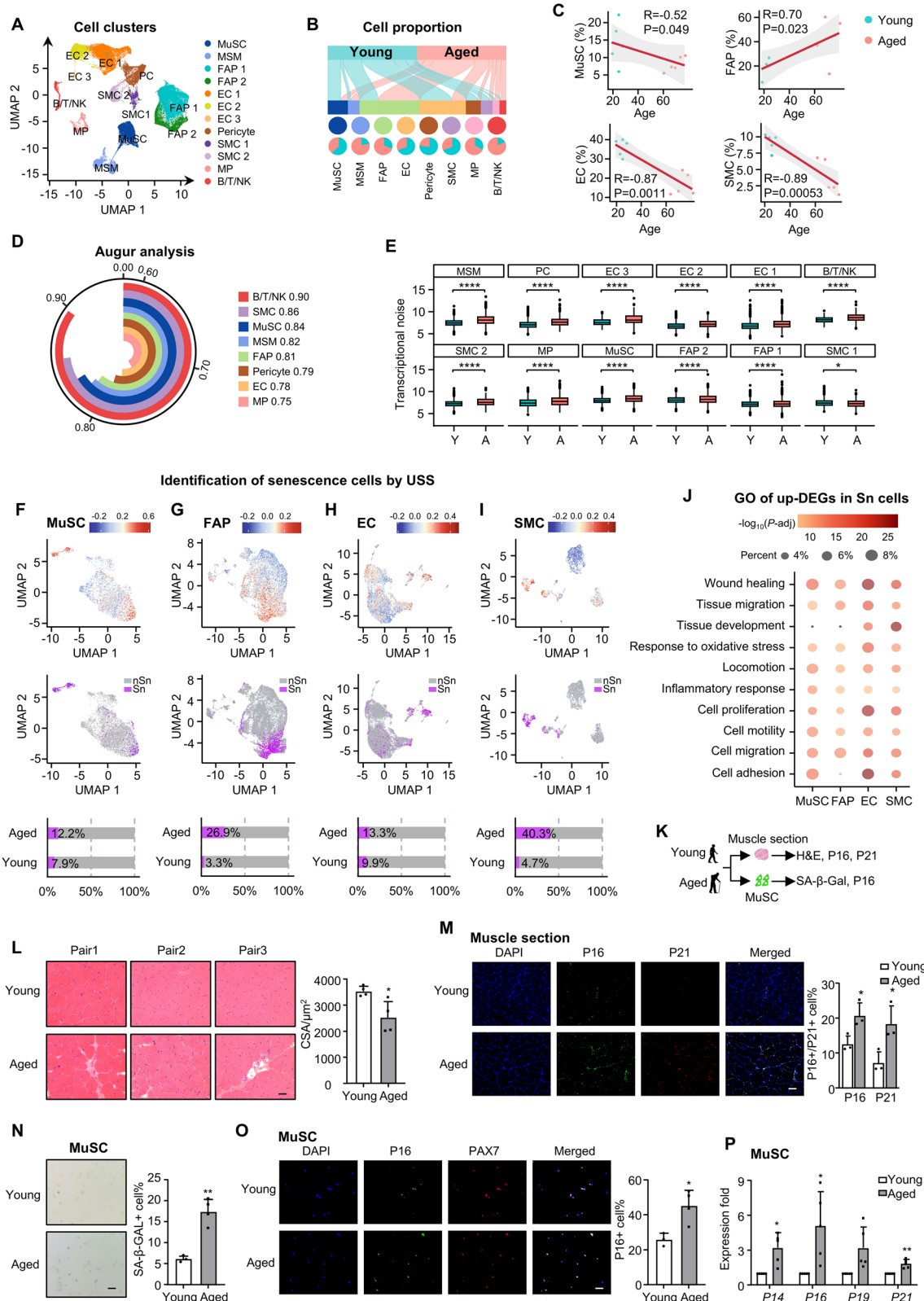

“Cytokine/chemokine activity” (such as *CCL2, CXCL12, CXCL8, CXCL2*), “Growth factor” (such as *FGF2, IGFBP6, IGFBP7*) etc. enriched in the late-phase MuSCs (Fig. 3F, Suppl. Fig. 2C). Moreover, the DEGs in the Late 1 and Late 2 branches were enriched for distinct GO terms with Late 1 being highly pro-fibrotic compared to Late 2 (Fig. 3G and Suppl. Dataset 3). Altogether, the above findings demonstrate the high level of heterogeneity in senescent MuSCs.

The pseudotime trajectories for other cell types were also examined. FAPs displayed a more continuous and smooth cell fate trajectory (Fig. 3H), along which the ratio of aged nuclei significantly increased (Suppl. Fig. 2D, E) and senescent FAPs accumulated near the end (Fig. 3I), accompanied by a decreased cell cycle ss-GSVA score (Fig. 3J). Inflammatory response genes such as *CDKN2B, CD56, CXCL8* and *CXCL1* were among the most enriched pseudotime-correlated DEGs in the late

**Fig. 2 | Multiomics mapping of senescence atlas in aging human muscle.**
**A** Uniform manifold approximation and projection (UMAP) plot showing the 12 (sub)types of muscle resident mononucleated cell populations. **B** Sankey plots showing the distribution and relative composition of young/aged cells across 8 main cell types. **C** Scatter plots of MuSC, FAP, EC, and MP proportions in young/ aged donors, with Pearson correlation coefficients (two-sided, unadjusted). The red regression lines represent the linear model fits, with the shaded areas indicating the 95% confidence intervals (CIs). **D** Arc plot showing the cell type aging responsiveness with augur scores from random forest (one-sided, unadjusted). **E** Boxplots of transcriptional noise in young (Y) vs. aged (A), ordered by A/Y ratio using Wilcoxon tests (two-sided, unadjusted). **F–I** Senescence atlas: UMAPs colored by SenMayo ssGSVA scores; senescent (Sn) vs. non-senescent (nSn) cells in MuSCs, FAPs, ECs, SMCs. Bar plots: Sn/nSn ratios in young/aged. **J** Dot plot of representative upregulated Sn DEG GO terms (hypergeometric test, one-sided; $p < 0.05$, Benjamini-Hochberg-adjusted). **K** Schematic of the experimental design for senescence detection in human muscle or freshly isolated MuSCs. Created in BioRender. Li, Y. (2025) https://BioRender.com/xsxka6w **L** H&E staining and quantification of cross-section areas (CSAs) of aged vs. young human muscle. Scale bar: 50 μm, $n = 4$. $p = 0.021$. **M** IF staining and quantification of P16 and P21 (red) on the above human muscle sections. Scale bar: 50 μm, $n = 3$. $p = 0.035$ (P16), 0.035 (P21). **N** SA-β-GAL staining and quantification of young/aged MuSCs. Scale bar: 50 μm, $n = 3$. $p = 0.0018$. **O** IF staining and quantification of P16 and PAX7 on the above MuSCs. Scale bar: 50 μm, $n = 3$. $p = 0.011$. **P** RT-qPCR detection of *P14, P16, P19* and *P21* genes expressions in the above MuSCs. $n = 5$. $p = 0.00020$ (*P14*), 0.037 (*P16*), 0.0070 (*P21*). All the bar graphs are presented as mean + SD, paired two-sided Student's *t*-test (**P**) and unpaired two-sided Student's *t*-test (D, L-O) were used to calculate the statistical significance: \**p < 0.05*, \*\**p < 0.01*, \*\**p < 0.001*, *n.s.* = no significance. Source data are provided as a Source Data file.

stage (Fig. 3K and Suppl. Fig. 2F, G). Similar to FAPs, ECs exhibited a relatively simple trajectory pattern, with cells starting from early-end, going through a middle branch, then falling into the late-end (Fig. 3L). Mapping of ECs along the trajectory also uncovered a significant increase of aged nuclei (Suppl. Fig. 2H, I), senescent ECs (Fig. 3M) and reduced cell cycle ss-GSVA score (Fig. 3N). Aging ECs were enriched for genes associated with "Extracellular space" (such as *COL1A2, CCDC80, PPKG1*) and "Transcription factor" (such as *ZEB2, MAP1B*) (Fig. 3O and Suppl. Fig. 2J, K). In the trajectory for SMCs, we observed a mixture of young and aged nuclei in the early-end and middle phases, whereas the late end was predominantly occupied with aged nuclei (Suppl. Fig. 2L, M). Most aged SMCs from the late end were detected as senescent and functionally distinct from the rest (Fig. 3P, Q). We also observed a consistent decrease in cell cycle ss-GSVA score along the SMC pseudotime axis (Fig. 3R); and pseudotime-associated genes were enriched for both "Inflammatory response" (such as *NFκB1, TIMP1, TNXB, TNFA1P6, RUNX1*) and "Extracellular space" (such as *THBS4, NOTCH2, ITGBL1*) (Fig. 3S; Suppl. Fig. 2N, O). Collectively, the findings reveal the temporal heterogeneity/dynamics of cellular senescence both inter- and intra-populationally among four major types of mononuclear cells in aged human muscle.

## SASP profiling and function in senescent cells

Knowing the importance of SASPs in determining the senescence heterogeneity and function, we next mapped the SASP dynamics in aging muscle. We first identified senescence-associated DEGs (Sn-DEG) comparing senescent (Sn) and non-senescent (nSn) cells across the four mononuclear cell populations. A total of 1514 up- and 1576 down-regulated Sn-DEGs were identified in at least one cell type and the majority were cell type specific (Fig. 4A and Suppl. Dataset 3), again reinforcing the inter-cellular heterogeneity. By intersecting with the SASP set, 243 of the SASPs were found up-regulated in senescent cells and 78 were commonly shared in at least two cell populations (Fig. 4B, Suppl. Dataset 4); in particular, 16 were commonly shared in all four cell types including *CXCL8, CXCL2, VCAN, COL12A1, MFAP5, PLAU, CXCL3, FBN1, CD44, CXCL1, SOD2, SLC39A14, MMP3, SERPINE2, ALCAM,* and *FSTL1* (Fig. 4C). Nevertheless, >30% of the SASPs were cell type specific (25, 36, 32, 48 in MuSC, FAP, EC and SMC) (Fig. 4B, Suppl. Fig. 3A); for example, *IGFBP7, NAMPT, CCL2, TGFB, IL7,* and *ANGPT2* were uniquely up-regulated in senescent MuSCs and *THBS1, MAT2A, GALNT2, DPP4, ITGB3, LRPPRC* in FAPs (Fig. 4D). The above findings demonstrate both the commonality and the inter-populational variation in SASP constitutions. When taking a close examination of the MuSC SASPs, expectedly, we found the SASP ssGSVA score was much higher in senescent vs. non-senescent MuSCs (Fig. 4E) and also in aged vs. young MuSCs (Fig. 4F). *DCN, VCAN, CXCL2, CCL2, CXCL1* were among the top-ranked SASPs in both senescent and aged MuSCs (Fig. 4G, H and Suppl. Dataset 4). The induction of *DCN, CXCL1, APOD, CCL2, CXCL2, CXCL8, IGFBP6,* and

*EGFR* was further validated by RT-qPCR in freshly isolated MuSCs (Fig. 4I). Altogether, the above results solidify the induction of SASPs and profile their dynamics in senescent mononuclear cells in aged human muscle.

Next, to elucidate the function of the SASPs, we mapped SASP-mediated intercellular communication, a total of 810 SASP-mediated ligand (L)-receptor (R) pairs were defined among all cell types by CellChat (Suppl. Dataset 4). Expectedly, SASP-mediated interaction strength was enhanced in the aged vs. young muscle (51.8 vs. 47.2, Fig. 4J) along with the overall L-R mediated cell-cell interaction strength (56.8 vs. 52.0, Suppl. Fig. 3B), suggesting SASP function in augmenting cellular communication and altering niche microenvironment in aging muscle. Consistently, further examination of the receiver-sender cell interactions revealed a global strengthening of SASP (Fig. 4K) or all L-R (Suppl. Fig. 3C) mediated interaction signals between two cell types; for example, the signals to SMCs or ECs from all other types of cells showed a remarkable increase in aged muscle. Interestingly, MuSCs received decreased signals from ECs, FAPs and Pericytes while sending higher levels of signals to BT/ NK cells, ECs, MPs and MSMs (Fig. 4K). Additionally, the network analysis revealed significantly altered SASP-mediated pathways in scaled information flow (the total sum of communication probability from the inferred SASP network); and a gain of inflammatory SASP-mediated pathways (for example by VISFATIN, CD226, ALCAM, and MHC-I signals) was observed in the aged muscle (Fig. 4L); ECM-SASP mediated pathways were also altered, for example, increased ITGB2 pathway and decreased PTN pathway were observed (Fig. 4L). Additionally, growth factors, such as FGF and IGF-mediated pathways were also changed in aged muscle (Fig. 4L). Of note, we found that CXCL family (CXCL12, 2, 8, 3 and 1) may act as key SASPs mediating cellular interactions and CXCL-mediated interaction strength and frequency showed significant elevation in the aged muscle (Fig. 4M, Suppl. Dataset 4). By examining enhanced signaling pairs in the aged group, we identified 54 SASP-mediated L-R pairs with MuSC as either a sender (43), receptor (9) or involved in autocrine interactions (2) (Fig. 4N, O, Suppl. Dataset 4). Among the most age-related increased signals emanating from or received in MuSCs, MPs, SMCs, and FAPs exhibited the highest interaction frequency with MuSCs; and CD44 represented a key receptor mediating the interactions (Fig. 4N, O). Next, to further examine the up-regulated SASP-mediated cell interactions in aged MuSCs, analysis of differential communication probability was performed to identify top-ranked L-R pairs emanating from MuSCs. For example, MIF-(CD74 + CXCR4) signaling from MuSCs to B/T/NK cells was drastically activated in aged muscle, and MIF-ACKR3 signaling from MuSCs to FAPs, FN1-CD44 signaling from MuSCs to SMCs were also increased (Fig. 4P). Altogether, the above findings define SASP-mediated cellular interactions and demonstrate the key functions of SASPs in modulating muscle microenvironment during muscle aging.

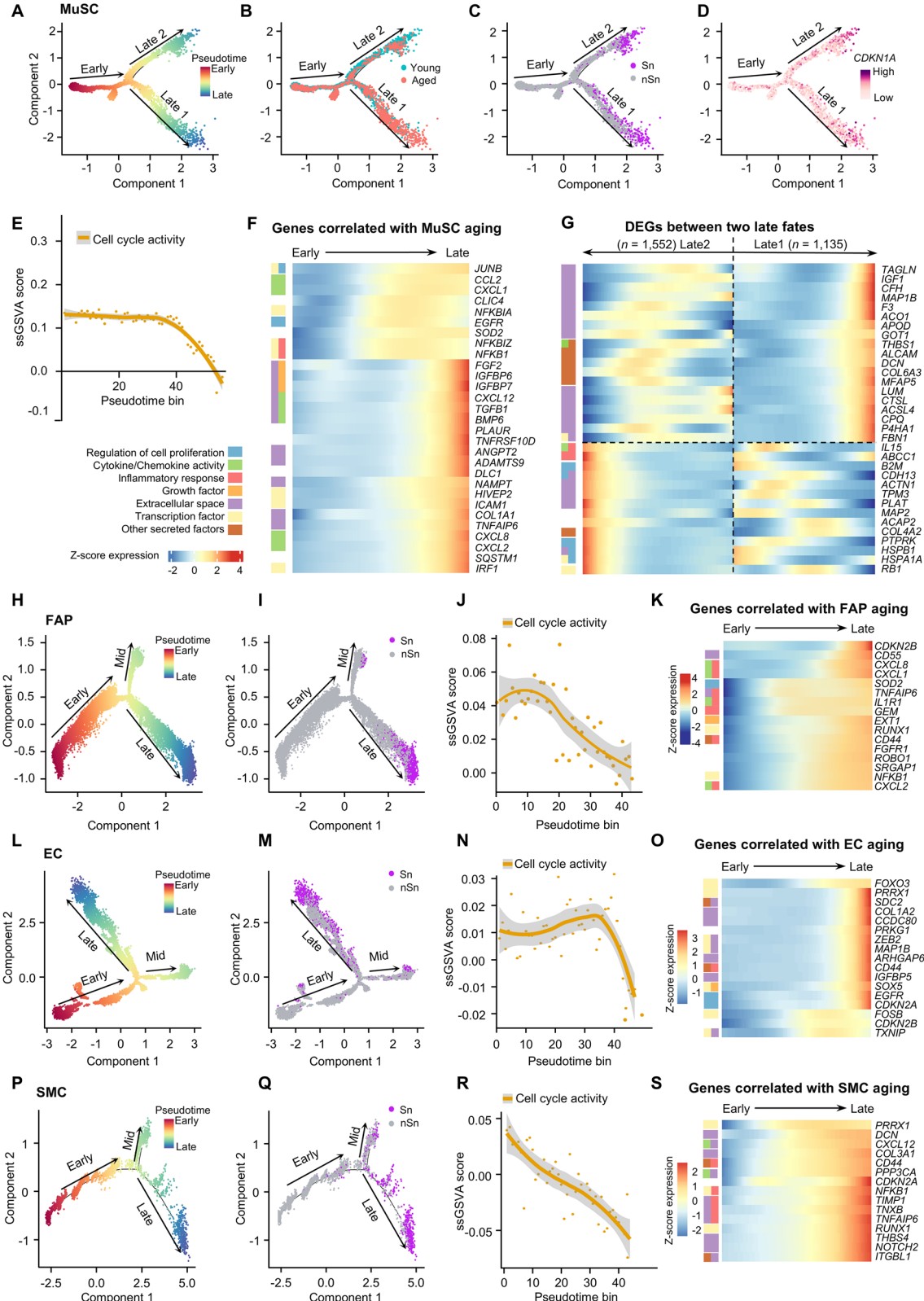

## Maraviroc as a potential senotherapeutic for sarcopenia

Among all the above-identified SASPs in senescent cells, we observed a robust induction of *CCL3, CCL4* and *CCL5* along with their receptor *CCR5* in aged vs. young MuSCs (Fig. 5A) and whole muscle (Suppl. Fig. 4A, B); and the high induction was also confirmed by RT-qPCR in isolated MuSCs from additional human muscle donors (Fig. 5B). Moreover, their induction was also found in aged mouse muscles and

MuSCs by analyzing publicly available single-cell RNA-seq data (Suppl. Fig. 4C, D) and RT-qPCR (Suppl. Fig. 4E)[36]. Interestingly, in our recent study[37], we have demonstrated that Maraviroc(MVC), a Ccr5 antagonist is very effective in targeting the Ccl5-Ccr5 axis thus mitigating inflammation in dystrophic mouse muscles. We thus tested if MVC can be a potential senomorphic for treating muscle aging by blocking the function of Ccl3, 4 and 5 in mice. In the first high dose short term

**Fig. 3 | Heterogeneity and dynamics of cellular senescence in aging human muscle. A–D** Discriminative dimensionality reduction (DDR) tree visualization of MuSC trajectory with mapping of pseudotime (**A**), age group (**B**), senescence annotation (**C**), and *CDKN1A* expression level (**D**). **E** Plot showing the ss-GSVA score of cell cycle activity signature genes along MuSC aging trajectory psuedotime. The solid yellow line is the local regression result for individual pseudotime bins (55 total, sized 0.10 per bin), with the gray shadow depicting the 95% CIs. **F** Heat map visualization of expression levels of genes (right) with correlated expression profiles to MuSC aging pseudotime ordered from Early to Late stage. Left: row annotation showing the functions of the genes. **G** Heat map visualization of expression levels of DEGs between Late 1 and Late 2 fates ordered by pseudotime. **H, I** DDR tree visualization of FAP trajectory with mapping of pseudotime (**H**) and senescence annotation (**I**). **J** Plot showing the ss-GSVA score of cell cycle activity signature

genes along FAP aging trajectory psuedotime with 95% CI. **K** Heat map visualization of expression levels of genes with correlated expression profiles to FAP aging pseudotime from Early to Late stage. **L, M** DDR tree visualization of EC trajectory with mapping of pseudotime (**L**) and senescence annotation (**M**). **N** Plot showing the ss-GSVA score of cell cycle activity signature genes along EC aging trajectory with 95% CI. **O** Heat map visualization of expression levels of genes with correlated expression profiles to EC aging pseudotime. **P, Q** DDR tree visualization of SMC trajectory with mapping of pseudotime (**P**) and senescence annotation (**Q**). **R** Plot showing the ss-GSVA score of cell cycle activity signature genes along SMC aging trajectory with 95% CIs. **S** Heat map visualization of expression levels of genes with correlated expression profiles to SMC aging pseudotime. Source data are provided as a Source Data file.

(HDST) treatment regime, a high dose (10 mg/kg) of MVC was administrated on 18 months old mice intraperitoneally for 3 months (Fig. 5C). The MVC treatment led to an evident increase in muscle mass (28.01% increase of TA/body weight) (Fig. 5D) and muscle morphology (Fig. 5E) compared to the DMSO-treated mice; by H&E staining, the inflammation was also attenuated, and the muscle fiber size was increased (15.50%) (Fig. 5E). As a result, the muscle function was significantly enhanced which was evidenced by a notable increase in the grip strength (15.79%) (Fig. 5F) and a tapered grip strength reduction (Δgrip) (19.75%) (Fig. 5G). Consistently, when subjected to a treadmill running test, in which the mice were adapted to a treadmill followed by a stepwise increase of running speed until their exhaustion, the MVC-treated mice demonstrated higher running speed (26.42%) (Fig. 5H) and longer running distance (50.00%) (Fig. 5I). Overall, we observed the mice were rejuvenated to a much healthier and active state. Furthermore, the number of MuSCs was elevated (3.91% vs. 3.39%) accompanied by decreased macrophages (0.89% vs. 1.09%) but no significant change of FAPs (Fig. 5J), indicating the treatment blocked the action of SASPs and improved the aging muscle niche.

The above-uncovered niche impact was further elucidated by single-cell RNA-seq analysis of the 5437 and 5454 mononuclear cells collected from DMSO or MVC-treated mouse muscles. Among the total of 12 identified cell populations including Pro-inflammatory MPs (PI-MPs), Anti-inflammatory MPs (AI-MPs), FAPs, EC1, EC2, Tenocytes, B/T/NK cells, SMCs, MuSCs, Neutrophils, MSMs, Pericytes and Schwann cells (Fig. 5K) based on normalized gene expression levels and canonical cell type specific markers (Suppl. Fig. 4F, and Suppl. Dataset 5), we detected a remarkably increased population of MuSCs in MVC vs. DMSO group (4.68% vs. 2.65%) accompanied by a reduced population of PI-MPs (1.91% vs. 3.79%) but interestingly not FAPs (27.72% vs. 23.28%) (Fig. 5L, Suppl. Dataset 5). Furthermore, we detected significantly decreased levels of cellular senescence in MuSCs (17.6% vs. 20.1%, Fig. 5M), FAPs (15.8% vs. 19.3%, Fig. 5N), ECs (16.7% vs. 21.9%, Fig. 5O) and SMCs (20.1% vs. 31.1%, Fig. 5P), which was further supported by the decreased *p21* mRNA expression (Fig. 5M–P). Consistently, reduced p16 and p21 expressions were detected in the whole muscles (Suppl. Fig. 4G, H) and also the isolated MuSCs after MVC treatment (Suppl. Fig. 4I). In addition, pseudotime analysis on the MuSCs uncovered that the cells progressed into one late fate after the MVC treatment but a middle branch emerged; indeed a lower portion of senescent cells distributed in the late fate confirming the decreased senescence after the MVC treatment (Suppl. Fig. 4J). Additionally, cellular crosstalk analysis revealed a global decline of cellular interactions in the MVC-treated muscle (Suppl. Fig. 4K, L). A close examination of SASP-mediated cellular interaction strength showed a decreased interaction strength (Fig. 5Q); further examination of the receiver-sender cell interactions also revealed a global decrease of SASP-mediated interaction signals between two cell types except for B/T/NK cell interactions (Fig. 5R). For instance, MVC treatment led to reduced interactions mediated by Cxcl pathway among MuSCs, SMCs, neutrophils, and MPs (Suppl. Fig. 4M). Expectedly, Ccr5 interactions

with its ligands, Ccl3, Ccl4 and Ccl5 were repressed by the MVC treatment (Fig. 5S, Suppl. Fig. 4N, O), which was also accompanied by their reduced expression levels in the whole muscle (Fig. 5T). To further examine the impact of MVC treatment on MuSCs, bulk RNA-seq was performed on freshly isolated MuSCs from the treated mice. We found that the 231 down-regulated genes (Fig. 5U, Suppl. Dataset 6) upon MVC treatment were enriched for SASPs-related terms such as "extracellular space", "inflammation response" etc. (Fig. 5V and Suppl. Dataset 6), suggesting repressed SASP expression; this was also confirmed by the Gene Set Enrichment Analysis (GSEA) (Fig. 5W and Suppl. Dataset 6). Expectedly, *Ccl3, Ccl4, Ccl5* and *Ccr5* were among the down-regulated SASPs (Suppl. Dataset 6), which was further confirmed by RT-qPCR (Fig. 5X). Altogether, the above results demonstrate the potential use of MVC as a senomorphic for alleviating cellular senescence, improving muscle niche and rejuvenating aging muscle.

To further demonstrate the efficacy of MVC treatment, we next tested a low dose (2 mg/kg) long-term (6 months) (LDLT) regime (Suppl. Fig. 5A) and found the treatment also led to a pronounced restoring effect of muscle morphology, function and niche integrity in aged mice (Suppl. Fig. 5B–G); we also tested a low dose (2 mg/kg) and short term (3 months) (LDST) regime (Suppl. Fig. 5H), and the treatment did not appear to have evident restoring effect: no significant changes in muscle morphology and muscle niche were observed despite increased TA weight and muscle performance (Suppl. Fig. 5I–N). Moreover, when the HDST regime was applied on 2 month-old young mice, no significant treatment effects were detected (Suppl. Fig. 5O–T), indicating the specificity of MVC efficacy on aged mice.

**Defining TFs governing senescence state and SASP induction in human muscle**

Next to gain a holistic understanding of the senescence state/SASP regulation and identify potential upstream TF regulators for senotherapeutic targeting, we analyzed the paired snATAC-seq data which permits identification of TF binding via mapping chromatin accessibility. As a result, key TF regulators of senescent cell state were defined with enriched motifs predicted in the ATAC measurements. 234 TFs were shared in at least two cell types, and 26 were commonly found in all four cell types, including NF-κB family (NF-κB1, REL and RELB), several AP-1 family TFs (ATF2, 3, 4, 6, 7 and BATF3), C/EBP family (C/EBPD, B, G, A) and CREB family (CREB3, 5 and CREB3L3, 4) (Fig. 6A). Both NF-κB1 and C/EBPB are known key players in the regulation of senescence and SASPs[7–11]. We thus took a close examination of the previously unappreciated ATF3 factor. Of note, the accessibility of ATF3 binding motifs was increasingly enriched in Sn vs. nSn cells (Fig. 6B) and also in aged vs. young cells (Fig. 6C) in all four cell types, demonstrating its potential role in regulating senescence and aging. By integrating the paired snRNA-seq and snATAC-seq data, Functional Inference of Gene Regulation (FigR) analysis[38] was performed to identify target genes that were activated or repressed by ATF3 binding (Suppl. Dataset 7). As a result, transcriptional scores of the ATF3-activated genes were notably elevated in Sn cells (Fig. 6D), while the

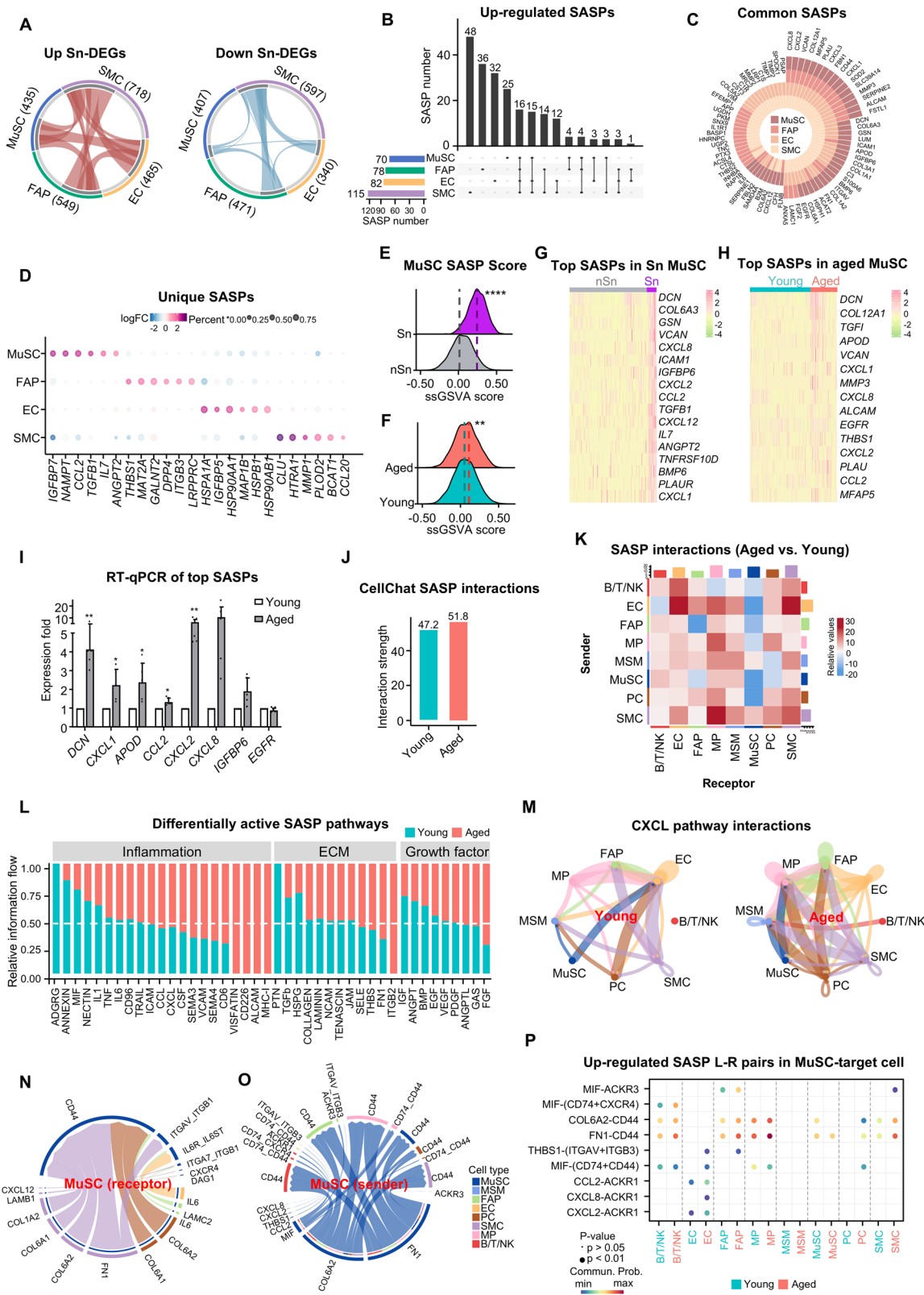

scores of the repressed genes were lower compared to the nSn group (Fig. 6E), supporting the notion that ATF3 plays a positive regulatory function in senescent cells. However, this difference was not pronounced in the comparison between the young and aged groups (Fig. 6D, E). Furthermore, GO functional analysis revealed that ATF3 target genes were enriched for senescence-related terms such as cell proliferation, cell migration, regulation of chemotaxis, regulation of

cell adhesion, and ECM organization, etc. (Fig. 6F and Suppl. Dataset 7). While displaying similar enrichment terms, differential enrichment patterns were observed in the four cell types. For instance, ATF3 target genes identified in MuSCs were highly enriched for the regulation of the MAPK cascade; target genes in FAPs exhibited a low association with cell morphogenesis while the targets in SMCs showed limited relevance to extracellular matrix organization. To construct the ATF3

**Fig. 4 | SASP profiling and function in senescent cells. A** Circos plots showing up/downregulated DEGs in senescent MuSCs, FAPs, ECs, SMCs (total DEGs in brackets). Curves showing shared Sn-DEGs between two cell types. Inner arcs (grey) showing unique/shared Sn-DEGs. **B** Upset plot showing the number of unique/shared upregulated SASP genes (pairwise cell comparisons). **C** Plot showing the up-regulated SASPs shared by at least two cell types. **D** Scatter Plot showing the cell type-specific upregulated SASPs. **E, F** Ridge map showing ss-GSVA score density for classical SASP genes in Sn vs. nSn (**E**) and aged vs. young (**F**). The dashed line corresponds to peak positions. $p = 2.2e{-}16$ (**E**), 0.0025 (**F**) (Wilcoxon, two-sided). **G, H** Heatmap of op upregulated SASPs in Sn vs. nSn (**G**) and aged vs. young (**H**). **I** RT-qPCR detection of top-ranked up-regulated SASP gene expressions in aged/young human MuSCs, $n = 5$. $p = 0.0043$ (*DCN*), 0.032 (*CXCL1*), 0.041 (*APOD*), 0.022 (*CCL2*), 0.0093 (*CXCL2*). **J** Bar plot comparing SASP-mediated intercellular interaction strength (aged vs. young). **K** Heatmap showing differential SASP interactions

(aged vs. young). Red/blue represents increased/decreased signaling. Top/right bars showing incoming/outgoing signals for each cell type. **L** Relative flows of differentially active signaling pathways during muscle aging. **M** Plot showing the signal strength change by aggregating all L-R pairs within CXCL pathway. The edge color corresponds to the sender cell type, and the edge weight is proportional to the interaction strength. **N, O** Chord diagram visualizing MuSC-centered SASP communication as receptor (**N**) or sender (**O**). **P** Dot plot showing the increased SASP L-R pairs from MuSC (sender cell) to other cells (target cell) in aged vs. young group. The dot color and size represent the computed communication probability and *p*-values. All the bar graphs are presented as mean + SD, paired two-sided Student's *t*-test (**I**) and unpaired two-sided Student's *t*-test (**E**, **F**) were used to calculate the statistical significance: *$p < 0.05$, **$p < 0.01$, ***$p < 0.001$, ****$p < 0.0001$, *n.s.* = no significance. Source data are provided as a Source Data file.

regulatory network, ATF3-regulated senescent DEGs (ATF3-SnTargets) were predicted and many were shared in multiple cell types (Fig. 6G and Suppl. Dataset 7). Interestingly, the shared target genes (such as genes *CXCL8, CXCL2*, and *CXCL1* shared in all four cell types) were predominantly upregulated in Sn vs. nSn cells (Fig. 6G), suggesting ATF3 mainly functions to promote gene expression in senescent cells. Altogether, the above findings define potential TF regulators of cellular senescence in aging muscle and highlight ATF3 as a previously unknown key regulator of senescence in multiple mononuclear cell populations in human muscle.

Next, we further elucidated the core TFs governing SASP induction, similarly we constructed TF-SASP regulatory networks and defined potential activating (Fig. 6H) and suppressing (Fig. 6I) TFs (Suppl. Dataset 7). Again, NF-κB (NF-κB1, REL), and AP-1 family (JUNB, FOSL1, ATF6, FOSB, JUND) TFs were the predominant activators in all four cell types (Fig. 6H); AP-1 family, such as JUNB, and FOSL1 were among the most probable TF activators (Fig. 6H). Interestingly, two other AP-1 family TFs, FOS and JUN, were identified as potential suppressors (Fig. 6I). To further explore the previously unappreciated JUNB-SASP regulation, we assessed JUNB-SASP association using regulation scores and defined 119 common or cell type specific JUNB-activated SASP targets (Suppl. Dataset 7); Among the top 20 targets, *PLAU* was commonly shared in all four cell types while *CXCL1, MMP2, CCL17*, and *BTD* were unique in MuSCs, FAPs, ECs and SMCs (Fig. 6J). To substantiate the potential role of JUNB in activating SASPs, DORC (domains of regulatory chromatin) analysis was performed following previous method[39,40] to correlate the accessibility of JUNB motif-containing peaks near each SASP target gene with its expression. The difference between the JUNB DORC chromatin accessibility and gene expression along the pseudotime axis was calculated; we found that for most of the JUNB-SASP targets, the chromatin accessibility gain preceded that of the expression change in MuSCs (Fig. 6K) as well as the other three cell types (Suppl. Fig. 6A–C). These observations were exemplified on the *FBN1* and *TNFRSF10D* genes during MuSC aging trajectory (Fig. S6D, E); for example, JUNB-related chromatin change of *FBN1* was identified as an early event, occurring within the first 47 pseudotime bins, preceding the RNA expression change. Altogether the above findings suggest JUNB could be a key upstream TF inducer of SASP production, warranting further investigation.

## JUNB activates SASP induction in senescent MuSCs via enhancer regulation

To further elucidate JUNB regulation of SASP induction and senescence in MuSCs, we quantified the TF-SASP association by regulation scores and confirmed that JUNB was among the most prominent candidate regulators for up-regulated SASPs (Fig. 7A). A closer examination revealed that the chromatin openness level of the snATAC-seq predicted JUNB binding sites was significantly increased in aged vs. young MuSCs (Fig. 7B), supporting the high probability for JUNB to be a key regulatory TF in aged MuSCs. Expectedly, *JUNB* expression itself

was also significantly elevated in aged MuSCs according to the accompanied scRNA-seq data (Fig. 7C). We also examined its levels in the isolated human MuSCs; despite AP-1 TFs can be quickly activated by the isolation procedure[41–43] in both young and aged MuSCs, a much higher level of JUNB was detected in aged MuSCs (133.35%, Fig. 7D). 54 SASPs were predicted to be bound by JUNB including the top-ranked *IL1R1, TGFB3, TNFRSF10D, TNFRSF1A, GDF15, PLAUR, CXCL1, PDGFB, CSF3, TIMP2* (Fig. 7A and Suppl. Dataset 7); which was also confirmed by RT-qPCR in aged human MuSCs (Fig. 7D). NF-κB family, on the other hand, regulated a very different set of SASPs (Fig. 7A). Interestingly, analysis of published scRNA-seq and scATAC-seq data from mice[44] also predicted mouse JunB (mJunB) as an upstream TF regulator of SASP induction in aged mouse MuSCs (Suppl. Fig. 7A); consistently, m*JunB* upregulation was also detected in the MuSCs freshly isolated from aged vs. young mice (58.10%); moreover, the above predicted human SASP targets of human JUNB (hJUNB) were also highly expressed in aged mouse MuSCs (Fig. 7E). These results suggested a conserved role of JUNB in regulating SASP induction in both human and mouse MuSCs. We thus leveraged an inducible MuSC-specific JunB knockout mouse (*JunB*-iKO) that was generated by crossing the *JunB* flox mouse with a *Pax7CreER; R26Yfp* mouse (Suppl. Fig. 7B–D) to further elucidate the regulatory mechanism. MuSCs were isolated from the Ctrl and iKO mice; expectedly we found the expression levels of the above-defined mJunB SASP targets were all significantly down-regulated (Fig. 7F). Moreover, over-expression of mJunB in the MuSCs isolated from young mice induced the expression of some SASP targets including *Il1r1, Plaur, Cxcl1* and *Timp2* (Fig. 7G). Interestingly, the above loss or gain of mJunB did not appear to affect the levels of SA-β-GAL and several marker genes such as *p16, p19, p21 and p53* (Suppl. Fig. 7E–K), suggesting mJunB can induce SASP target activation but may not be sufficient to trigger full senescence program in mouse MuSCs.

To further explore how hJUNB or mJunB activates SASP genes, we took a close examination of the snATAC-seq data and found a large percentage of the predicted hJUNB binding was mapped to intergenic and intron regions (56.31% in young and 35.59% in aged) in MuSCs (Fig. 7H), indicating hJUNB may regulate gene expression mainly through enhancer binding. To further dissect the regulatory mechanism, CUT&RUN-seq was conducted to map mJunB binding in MuSCs from young and aged mice to define a total of 13,437 and 14,123 mJunB binding events (Suppl. Fig. 7L and Suppl. Dataset 8). Consistent with the above prediction in human MuSCs, a large percentage of mJunB binding peaks were mapped to intergenic and intron regions (63.50% in young and 58.99% in aged) in MuSCs (Fig. 7I), and >60% of the binding sites were >3 kb distal to the TSSs while a small portion of promoter binding (≤3 kb) was observed (36.66% in aged and 31.92% in young) (Suppl. Fig. 7M). By intersecting with the H3K27ac CUT&RUN-seq data performed in mouse MuSCs, we found indeed most mJunB-binding sites were in active enhancer regions (Fig. 7J), and its enhancer binding evidently increased in aged vs. young MuSCs (6,648 vs. 6,042, Suppl. Fig. 7N). Based on promoter and enhancer binding, we

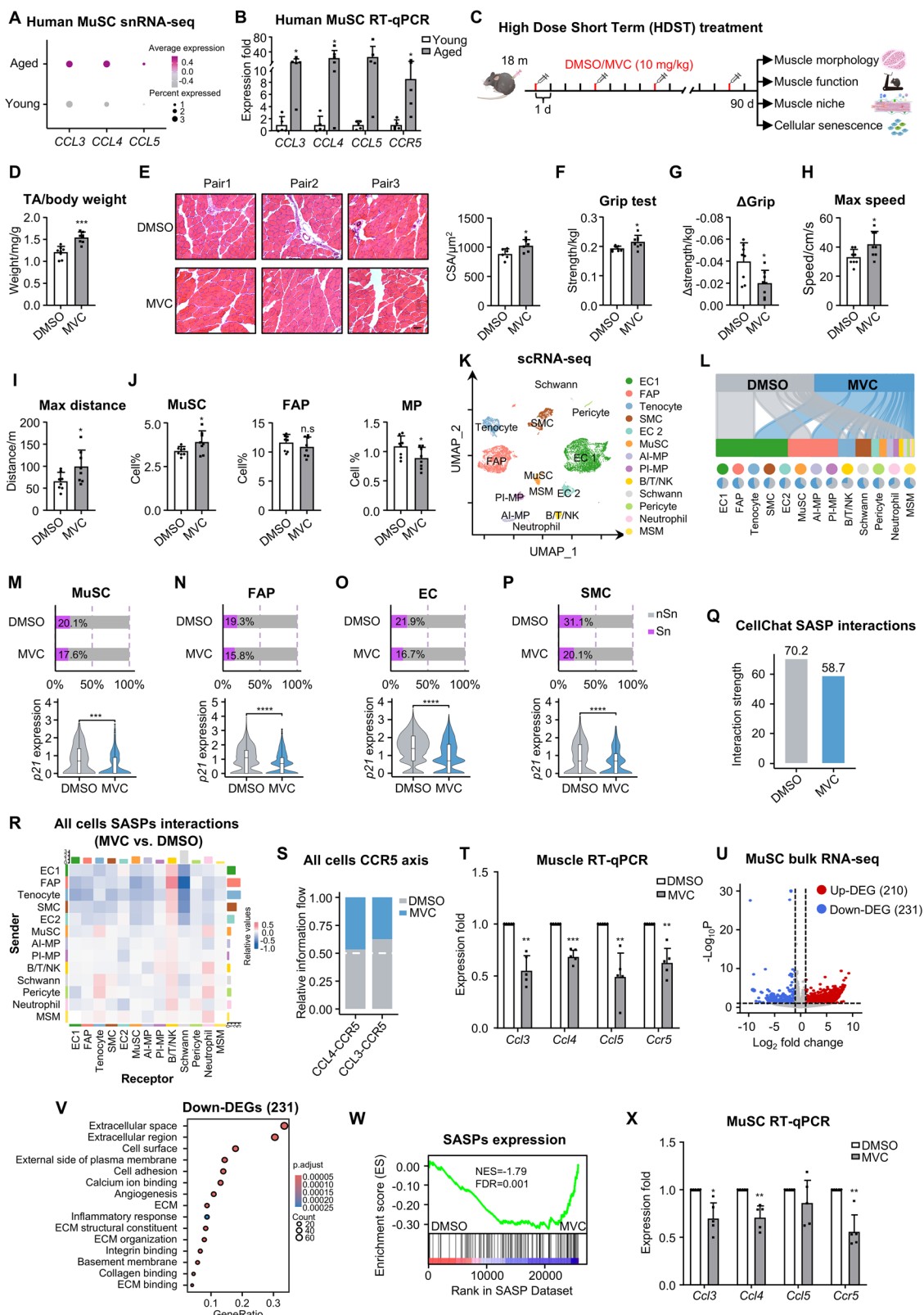

identified a total of 7,361 mJunB target genes in young and 10,653 in aged MuSCs (Fig. 7K and Suppl. Dataset 8), which were enriched for GO functions related to "protein modification" and "catabolic process" (Suppl. Fig. 7O and Suppl. Dataset 8). To elucidate how mJunB activates SASP transcription, we defined a number of the SASP targets that were directly bound by mJunB (Suppl. Dataset 8), and the number indeed increased in aged (40) vs. young (27) MuSCs (Fig. 7L). Expectedly,

mJunB binding mostly resided in the enhancer regions of these SASP targets. Notably, *Cxcl1* was defined as a prominent target in both human and mouse (Fig. 7D, E) and mJunB bound to both the promoter and an enhancer region of *Cxcl1* in aged mouse MuSCs (Fig. 7M). In human MuSCs, the predicted hJUNB binding on *CXCL1* resided in a downstream region and showed a significantly elevated chromatin openness level in senescent (Fig. 7N) and aged human MuSCs (Fig. 7O).

**Fig. 5 | Maraviroc is a potential senotherapeutic for sarcopenia. A** snRNA-seq analysis of *CCR5* axis gene expressions in human MuSCs. **B** RT-qPCR detection of *CCR5* axis gene expressions in human MuSCs, *n* = 5. *p* = 0.012 (*CCL3*), 0.039 (*CCL4*), 0.026 (*CCR5*). **C** Schematic of high dose short term DMSO/MVC treatment/ assessment regime in aging mice, *n* = 8. Created in BioRender. Li, Y. (2025) https:// BioRender.com/xsxka6w **D** The ratio of TA muscle/body weight of the above-treated mice, n = 8. *p* = 0.00040. **E** H&E staining and quantification of CSAs of TA muscles collected from the above-treated mice. Scale bar: 50 μm, *n* = 6. *p* = 0.034. **F, G** Grip strength and strength changes of the above-treated mice, *n* = 8. *p* = 0.014 (F), 0.017 (G). **H, I** Maximal running speed and distance of the above-treated mice, *n* = 8. *p* = 0.031 (**H**), 0.043 (**I**). **J** Flow cytometry detection of MuSC, MP, and FAP populations in the above-treated mice, *n* = 8. *p* = 0.045 (MuSC), 0.042 (MP). **K** scRNA-seq of mononucleated muscle cells isolated from the treated mice. Unsupervised clustering resolved 13 cell types. **L** Sankey plots showing the distribution and relative composition across cell types. **M**–**P** Top: Bar plot of USS-defined Sn/nSn cells. Bottom: Violin plot of p21 expression. *p* = 8.07e-05 (MuSC),

7.43e-24 (FAP), *p* = 1.31e-67 (EC), *p* = 4.99e-06 (SMC). Wilcoxon test (two-sided, unadjusted). **Q** Bar plot showing SASP-mediated interaction strength in DMSO and MVC. **R** Heatmap showing the altered SASP-mediated cell-cell interaction of MVC vs. DMSO. **S** Interaction frequency of Ccl3-Ccr5/Ccl4-Ccr5 pairs in MVC vs. DMSO. **T** RT-qPCR detection of the *Ccr5* axis gene expressions in whole muscles of MVC vs. DMSO, *n* = 5. *p* = 0.0024 (*Ccl3*), 0.00048 (*Ccl4*), 0.0084 (*Ccl5*), 0.0047 (*Ccr5*). **U** Volcano plot displaying DEGs from bulk RNA-seq performed in MuSCs from DMSO/MVC-treated mice. Log2FC >1, adjusted-*p* <0.05 (two-sided Wald test, Benjamini-Hochberg adjusted). **V** GO analysis of the above-identified 231 down-regulated DEGs. **W** GSEA analysis of the repressed SASPs in the above MVC vs. DMSO-treated MuSCs. **X** RT-qPCR detection of the *Ccr5* axis gene expressions in the above MuSCs, *n* = 5. *p* = 0.017 (*Ccl3*), 0.0074 (*Ccl4*), 0.0058 (*Ccr5*). All the bar graphs are presented as mean + SD, paired two-sided Student's *t*-test (T, X) and unpaired two-sided Student's *t*-test (**B**, **D**–**J**) were used to calculate the statistical significance: **p* <0.05, ***p* <0.01, ****p* <0.001, *n.s.* = no significance. Source data are provided as a Source Data file.

Furthermore, DORC analysis on *CXCL1* locus illustrated the chromatin accessibility gain preceded the transcriptional induction, indicating the activating role of hJUNB in *CXCL1* transcription (Fig. 7P). Altogether, the above results solidify the key role of hJUNB/mJunB in governing SASP induction in aged MuSCs.

## Discussion

In this study, we conducted multiomics mapping of cellular senescence atlas in aging human muscle. Using a USS scoring system, we mapped the cellular senescence in the mononucleated cells in aging muscle and uncovered commonality and heterogeneity of senescence state among different cells. Knowing SASPs are the main determinant of senescence state and function, we further defined SASP composition and revealed the heterogeneity and dynamics in SASP constitution and expression. Moreover, we dissected key TFs governing cellular senescence and SASP production and defined the key role of AP-1 family TF such as ATF3 and JUNB in senescence/SASP regulation. Our study solidifies the prevalence of senescence in multiple mononuclear cells and pinpoints the presence of MuSC senescence and its potential function in altering niche microenvironment. More importantly, the above mapping led to the identification of MVC as a potential senotherapeutic approach to delay sarcopenia progression and rejuvenate aging mice.

Despite the renewed interest in studying cellular senescence and its implication in organismal aging, there is still a scarcity of knowledge in our understanding of cellular senescence in aging skeletal muscle. Our study provides the first comprehensive mapping and characterization of cellular senescence in aging human muscle. First, this is the first multiomics mapping harnessing simultaneous single-nucleus RNA-seq/ATAC-seq to enable dissection of transcriptomic and epigenomic features of senescence in the same cells. By removing the terminally differentiated, post-mitotic myofibers from the analysis, it allowed us to focus on the mononucleated cells in the niche that are prone to senesce thus permitting an in-depth examination of senescence in MuSCs for the first time. Moreover, our study yields the first senescence blueprint on skeletal muscles from human. Although two recently emerged studies[18,36] provide the first cell atlas in the aging human skeletal muscles, none focus on cellular senescence. Lastly, our mapping is largely facilitated by creating a USS scoring method that combines almost all the known senescent gene sets, which permits a useful tool for defining the wide spectrum of senescence. The heterogeneous nature of senescence across cells and tissues is a well-recognized feature that hurdles the overall advancement of the field[4]. A holistic understanding of the transcriptomic and epigenomic heterogeneity of senescent cells in aging human muscles will contribute to a better grasp of the biological functions of these cells, and also facilitate the identification of new markers and therapeutic strategies for alleviating sarcopenia and

promoting healthy aging. Our mapping indeed uncovered wide variations of senescence at both inter- and intra-populational levels in aging muscle. Pseudotime analysis uncovered a temporal dynamic of senescence within each of the four examined cell types; senescent MuSCs clearly diverged into two different fates at the intra-populational level and senescence also showed marked different temporal patterns within FAPs, ECs and SMCs. The heterogeneity is also largely reflected by the SASP constitutions; unique compositions of SASP factors were defined in each cell population. Still, we were able to uncover shared features of senescence among the cells and common SASPs were found (Fig. 4); these SASPs constitute promising clinical biomarkers for assessing the burden of senescence in aging muscle tissue and also can serve as senomorphic targets for simultaneously blocking SASP action in multiple cells.

Functionally SASP factors are the major mediators of the non-cell-autonomous effects of senescent cells through their paracrine effect in dispersing senescence and altering the niche microenvironment[5,6]. Indeed, the cell-cell interaction analysis sheds light on the complex SASP-mediated cellular interactions. Of note, inflammatory SASP-mediated pathways were largely enhanced in aged muscle (Fig. 4L), supporting the key role of SASPs in inflammaging occurring in the aged skeletal muscle tissue. These defined interactions such as CXCL-mediated interactions occurring among multiple cell types thus represent potential targets for senomorphic design. Notably, the in-depth analysis of MuSC-centered cellular interactions reinforced our belief that MuSCs are not merely passive recipient of niche signals, they can actively modulate the niche through their secretory function[37]; here in aged human muscle, our finding defined many signaling pairs through which MuSCs communicate with other cell populations via secreting SASPs (Fig. 4N–P).

Lastly, the heterogeneous nature of SASPs/senescence can also arise from the inducers and regulators of SASPs which are relatively less characterized[5,6]. By harnessing the snATAC-seq dataset we gained the first comprehensive mapping of the upstream TF regulators of both senescence state and SASPs in aging human muscle. Along with NF-κB, the well-characterized master regulator of SASPs, AP-1 family appeared as a dominant transcriptional activator of senescence/SASPs in multiple cell populations. AP-1 family TFs are known to mediate early stress responses[45–47] and we recently elucidated the important role of ATF3 in regulating MuSC regenerative activity[48]. The findings from the current study illuminated a key role of ATF3 in governing senescent state in aging muscle via its extensive regulatory targets/networks, which warrants further dissection in the future. In terms of SASP regulation, we identified JUNB, also an AP-1 factor, as a prominent activator of SASP transcription in multiple cells and conducted in-depth mechanistic dissection on how JUNB activates SASPs in MuSCs. Our findings demonstrate that JUNB activates SASP activation in both human and mouse MuSCs. The direct JUNB SASP targets were

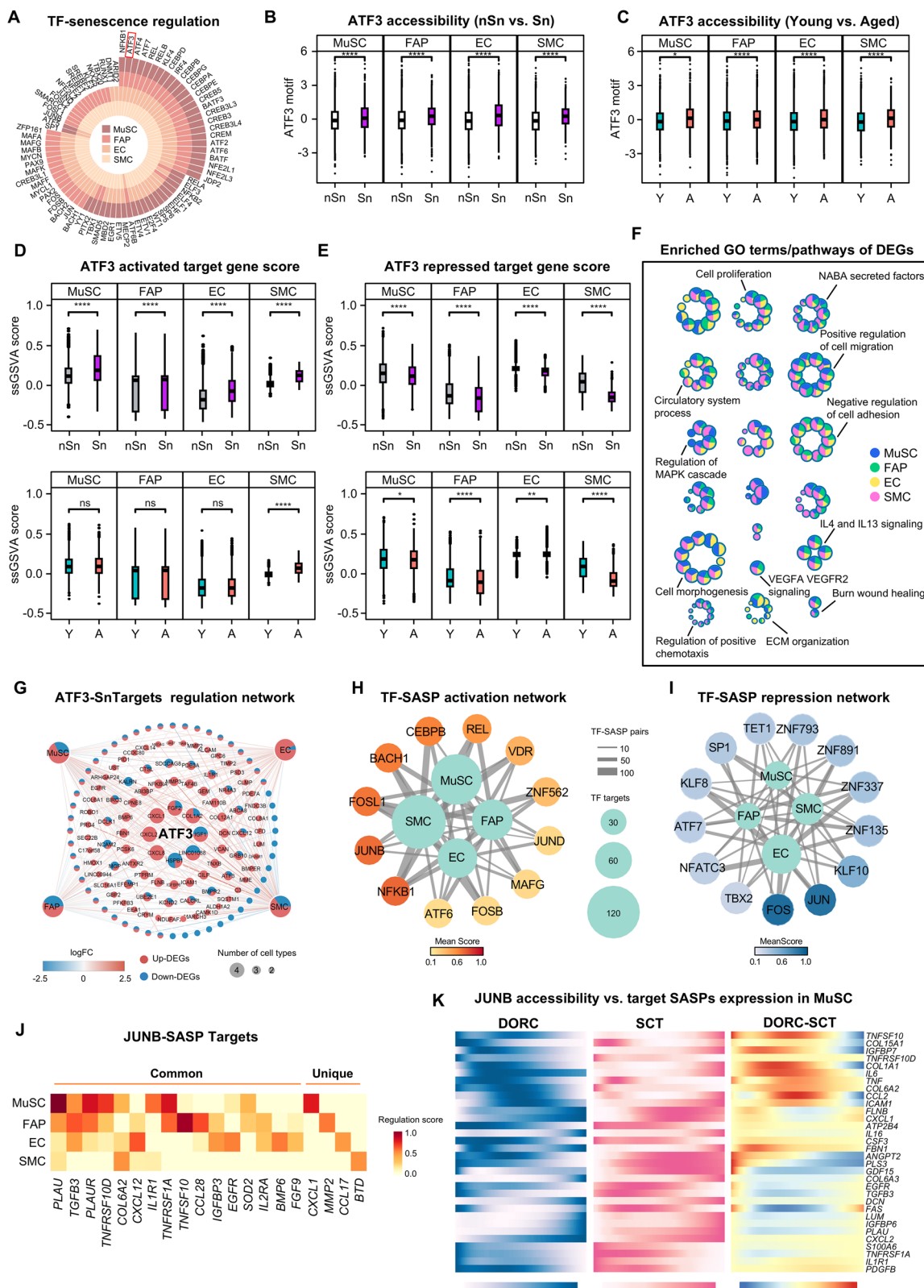

identified and CXCL1 appeared to be a prominent one shared in both human and mouse. Considering AP-1 TFs have been noticed as a possible regulator of senescence state in multiple cells/tissues[49–54], we thus reason targeting JUNB or other AP-1 TFs could be a powerful senomorphic approach that is worth further investigation in the future.

Collectively, we believe senescence heterogeneity encompasses multiple layers, by harnessing snRNA-seq/snATAC-seq our findings highlight the transcriptomic and epigenomic variations at intra- and inter-populational levels in aging human muscle. In the future, further advancement in single-cell technology will permit multi-modal characterization of both common and heterogeneous features of SnCs in aging muscle such as secretome, proteome, metabolism, etc. In addition, it will also be necessary to include immune cells in future endeavors which were excluded from the current study due to the

**Fig. 6 | Defining TFs governing senescence state and SASP induction in human muscle. A** Plots showing the predicted TFs governing senescence state shared by at least two cell types. **B, C** Box plots showing predicted ATF3 ATAC accessibility level in Sn vs. nSn cells (**B**) or aged vs. young groups (**C**). Exact *p*-value and other statistical details are provided in Source Data file. **D, E** Box plots showing the ss-GSVA gene set scores of target genes activated (**D**) or repressed (**E**) by ATF3. Full statistical details are provided in the Source Data file. **F** Network visualization of representative GO terms and pathways of ATF3-modulated DEGs in each cell type of aged vs. young muscle. The nodes represent GO terms or pathways, and the pie plots display the proportion of genes corresponding to a specific GO term or pathway in each cell type. **G** Network visualization of ATF3 targeted up- or down-regulated senescent genes in each cell type. Node size positively correlates with the number of cell types with its embedded pie chart indicating the number of up- and down-

regulated DEGs. Each connecting line represents Sn-DEGs in the corresponding cell type with its color indicating log2fold change (FC) values. **H, I** Network visualization of core activators (**H**) or repressors (**I**) TFs in each cell type between old and young groups. Outer nodes display different cell types and the node color represents the regulation score of all TF-SASP associations averaged on each cell type. Inner nodes positively correlate with the number of all TF-SASP pairs for each cell type. Each connecting line represents the number of SASP factors regulated by certain TF for each cell type. **J** Heatmap showing JUNB-SASP regulation score for each cell type. **K** Heatmaps highlighting smoothed normalized JUNB DORC accessibility, SCT-normalized RNA expression, and DORC-RNA difference for JUNB target SASP genes in MuSCs (*n* = 32). Two-sided unpaired Student's *t*-test was used to calculate the statistical significance (**B–E**): \**p* <0.05, \*\**p* <0.01, \*\*\**p* <0.001, \*\*\*\**p* <0.0001, *n.s.* = no significance. Source data are provided as a Source Data file.

challenge in discerning between their non-senescent and senescent states using the inflammatory SASPs.

Ultimately, it is our goal to identify druggable targets for senotherapeutics to ameliorate sarcopenia and enhance healthy muscle aging. Our attempt to use MVC in aging mice yielded very positive results. Systemic delivery of MVC that blocked CCL3, 4, 5 action significantly enhanced muscle performance, delayed muscle aging and led to remarkable rejuvenating effects; the effect was pronounced under the HDST, LDLT treatment schemes and milder under LDLT. The treatment diminished senescence in multiple cells and reversed the deregulated aging muscle niche, altogether demonstrating the feasibility of pharmacological SASP inhibition as a potential senomorphic therapy for sarcopenia and potentially other conditions involving CCLs producing senescent cells. Currently, there is very limited research regarding senotherapeutics and their impact on skeletal muscle in the context of chronological aging[12], our findings thus demonstrate the utility of MVC for ameliorating muscle aging. Recently, we also demonstrated the anti-inflammatory effect of MVC treatment on DMD muscle[37], therefore it could represent a potential pharmacologic intervention that concurrently targets both cellular senescence and inflammation, the two intricately associated hallmarks of organismal aging[1]. In the near future, it will also be possible to conduct a focused screen for senolytic or senomorphic compounds that target the unique features of senescent cells in skeletal muscle based on the foundational knowledge provided in our current study.

## Methods

### Human muscle biopsy and ethical clearance

Hamstring muscle samples were collected during orthopaedic surgery and informed consents were obtained in written form from the patients in the Hong Kong cohort, with ages between 19 and 27 years old (young group undergoing anterior cruciate ligament reconstruction) and 60–77 years old (aged group undergoing knee replacement). Structurally intact and histologically healthy hamstring muscles were collected from anatomically equivalent regions across cohorts. The informed consent was obtained from the legally acceptable representative. Ethical approval was granted by the Joint Chinese University of Hong Kong-New Territories East Cluster Clinical Research Ethics Committee (2021.255-T). Exclusion criteria were myopathy, hemiplegia or hemiparesis, rheumatoid arthritis or other autoimmune connective tissue disorders, cancer, coronary heart disease, inability to consent, or major surgery in the previous 3 months. Full metadata information for the organ donors is provided in Supplementary Dataset 1.

### Mice

All animal handling procedures, protocols and experiments ethics approval were granted by the CUHK AEEC (Animal Experimentation

Ethics Committee) under Ref No. 22-292-HMF. The mice were maintained in an animal room with 12 h light/12 h dark cycles, temperature (22–24 °C), and humidity (40–60%) at the animal facility in CUHK, fed with PicoLab® Select Mouse Diet 50 IF/9 F Diet and provided with plenty of fresh clean water at all times. For all animal-based experiments, at least three pairs of littermates or age-matched mice were used. All animals were euthanized by asphyxiation before experiments.

C57BL/6 aged mice were purchased from Gempharmatech (Nanjing, Jiangsu, China). JunB^f/f mouse strains were purchased from The Jackson Laboratory (Bar Harbor, ME, USA). The JunB-inducible conditional KO (iKO, *Pax7^CreERT2/R26Yfp*; *JunB^f/f*) strain and Ctrl (*Pax7^CreERT2/R26Yfp*; *JunB^+/+*) mice were generated by crossing *Pax7^CreERT2/R26Yfp* mice with *Junb^f/f* mice. Primers used for genotyping are shown in Suppl. Dataset 9.

### Animal procedures

Inducible deletion of JunB was administered by intraperitoneal (IP) injection of tamoxifen (TMX) (Sigma-Aldrich, T5648) at 100 mg/kg (body weight). Maraviroc (Sigma-Aldrich, PZ0002-25MG) treatment in aged C57BL/6 mice was administered by IP injection at 2 mg/kg (low dosage) or 10 mg/kg (high dosage) every 2 days for 3 months (short term) or 6 months (long term). For grip strength test, limb muscle grip strength of mice was measured by a grip strength meter (Kewbasis, KW-ZL-1) 3 times, the average values were calculated. For treadmill test, mice were adapted to a treadmill (Panlab, Harvard Apparatus, 76-0895) with a 5° incline at an initial speed of 10 cm/s, followed by a stepwise increase of 5 cm/s every two min until their exhaustion.

### Fluorescence-activated MuSC sorting and culturing

Muscle stem cells, fibro-adipogenic progenitors and macrophages were sorted based on established method[41–43,55–57]. Briefly, hindlimb muscles from mice and hamstring muscles from humans were digested with collagenase II (LS004177, Worthington, 1000 units per 1 ml) for 90 min at 37 °C, the digested muscles were then washed in washing medium (Ham's F-10 medium (N6635, Sigma) containing 10% horse serum, heat-inactivated (HIHS, 26050088, Gibco, 1% P/S) before cells were liberated by treating with Collagenase II (100 units per 1 ml) and Dispase (17105-041, Gibco, 1.1 unit per 1 ml) for 30 min. The suspensions were passed through a 20 G needle to release cells. Mononuclear cells were filtered with a 40 µm cell strainer and sorted by BD FACSAria IV with the selection of the GFP+ (MuSCs of Ctrl and Junb iKO mice); FITC-(CD45-, CD31-) APC-(SCA1-) PE+ (VCAM + ) (MuSCs of young and aged mice); FITC-(CD45-, CD31-, CD34-) APC + (CD29 + ) PE-CY7+ (CD56 + ) (MuSCs of human); FITC-(CD45-, CD31-, ITGA7-) APC + (SCA1 + ) (FAPs); FITC-(Cd45-) APC-(Ly6G-) eFluor450 + (CD11b + ) (MPs). Flowjo V10.8.1 was used for analysis of flow cytometry data. MuSCs were cultured in Ham's F10 medium with 20% FBS, 5 ng/ml β-FGF (PHG0026, Thermo Fisher Scientific) and 1% P/S, on coverslips and culture wells which were coated with poly-D-

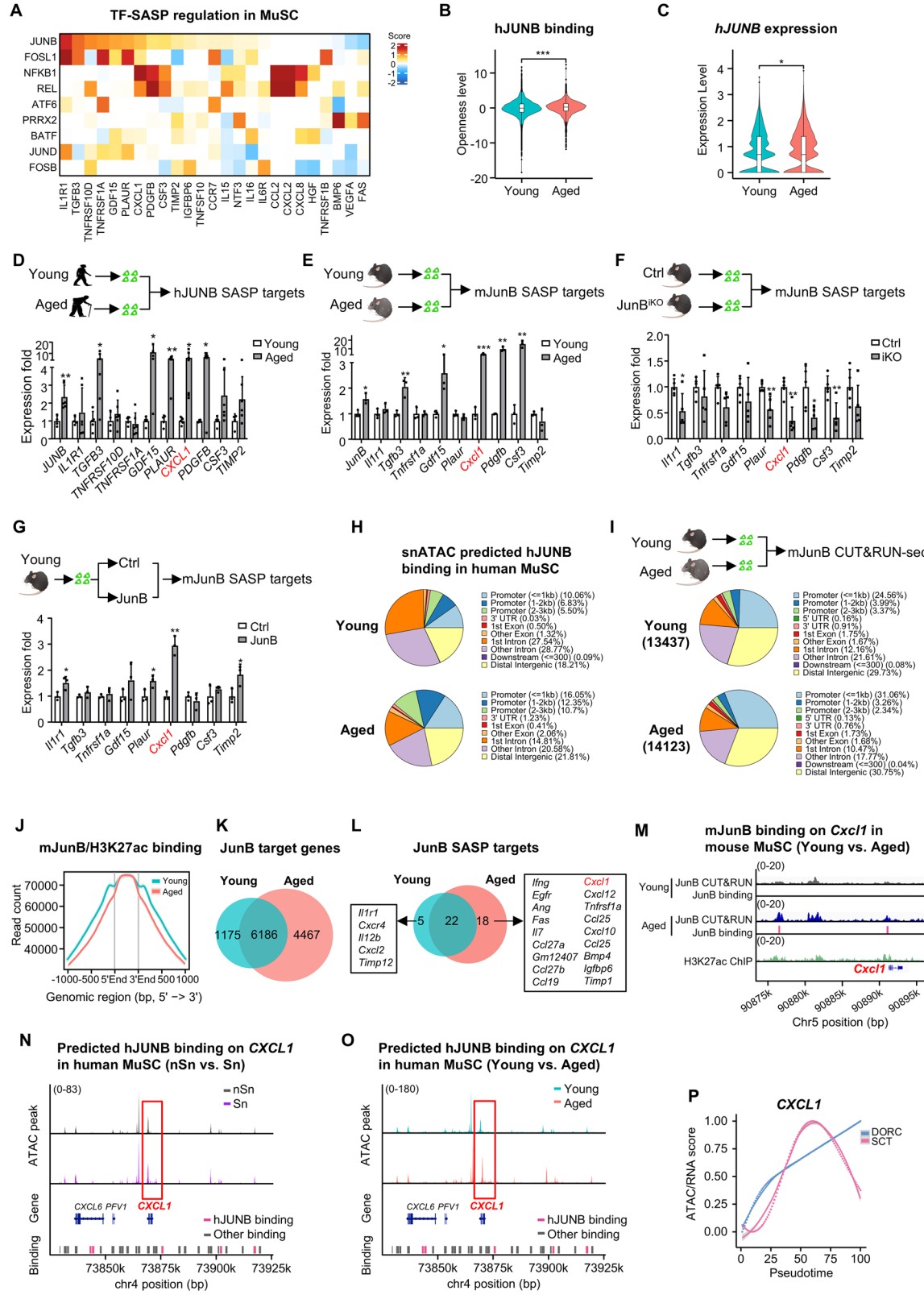

## Plasmids
pcDNA3.1-mouse-JunB plasmids were purchased from Youbio (http://www.youbio.cn/).

## RNA extraction and real-time PCR
Total RNAs were extracted using TRIzol reagent (Invitrogen) following the manufacturer's protocol. For quantitative RT-PCR, cDNAs were reverse transcribed using HiScript III First-Strand cDNA Synthesis Kit (Vazyme, R312-01). Real-time PCR reactions were performed on a LightCycler 480 Instrument II (Roche Life Science) using Luna

Also visible before these sections:
lysine solution (p0899, Sigma) at 37 °C overnight and then coated with extracellular matrix (ECM) (E-1270, Sigma) at 4 °C for at least 6 h.

**Fig. 7 | JUNB activates SASP induction in senescent MuSCs via enhancer regulation. A** Heatmap showing snATAC-seq DORC regulation scores for top TF-SASP associations in human MuSCs. **B, C** Violin plot showing chromatin accessibility (**B**) and expression (**C**) of human (*h)JUNB* in young vs. aged MuSCs. $p = 0.00032$ (**B**), 0.038 (**C**). Wilcoxon tests, two-sided. **D** RT-qPCR detection of the *hJUNB* and SASP target expressions in young/aged MuSCs, $n = 4$. $p = 0.0067$ (*JUNB*), 0.044 (*TGFB3*), 0.033 (*GDF15*), 0.0018 (*PLAUR*), 0.039 (*CXCL1*), 0.011 (*PDGFB*). **E** RT-qPCR detection of the *mouse (m)JunB* and SASP target expressions in young (2 m)/aged(24 m) MuSCs, $n = 3$. $p = 0.016$ (*JunB*), 0.0087 (*Tgfb3*), 0.034 (*Gdf15*), 0.0048 (*Cxcl1*), 0.0044 (*Pdgfb*), 0.0029 (*Csf3*). **F** RT-qPCR detection of the *mJunB* and SASP targets expressions in Ctrl/JunB-iKO MuSCs, $n = 5$. $p = 0.023$ (*Il1r1*), 0.0072 (*Plaur*), 0.0017 (*Cxcl1*), 0.0020 (*Pdgfb*), 0.0055 (*Csf3*). **G** RT-qPCR detection of the *mJunB* and SASP target expressions in Ctrl/JunB-overexpressing MuSCs, $n = 3$. $p = 0.037$ (*Il1r1*), 0.041 (*Plaur*), 0.0012 (*Cxcl1*), 0.041 (*Timp2*). **H** Pie chart showing the distribution of snATAC-seq predicted hJUNB binding sites in young/aged human MuSCs. **I** Pie chart showing the distribution of mJunB binding in young/aged MuSCs, detected by CUT&RUN-seq. **J** Average H3K27ac signal profile ( ± 1 kb around mJunB sites). **K, L** Pie chart showing the overlapping of mJunB target genes (**K**) and SASP genes (**L**) (promoter and enhancer bound) in young/aged MuSCs. The young/aged unique SASP targets are listed. **M** Genomic snapshots on *Cxcl1* locus showing the binding peaks of mJunB and H3K27ac in young/aged MuSCs. **N, O** Genomic coverage plot on *CXCL1* locus showing the predicted hJUNB binding peaks in Sn vs. nSn (**N**) and young vs. aged (**O**) human MuSCs. **P** Chromatin openness level (DORC) versus normalized gene expression (SCT) dynamics of *CXCL1* gene along MuSC aging pseudotime. Dotted line represents LOESS fit to the values obtained from sliding window bin averaged from DORC accessibility or SCT expression levels ($n = 100$ cells per bin). All the bar graphs are presented as mean + SD, two-sided paired Student's *t*-test was used to calculate the statistical significance (**B**–**G**): *$p < 0.05$, **$p < 0.01$, ***$p < 0.001$, n.s. = no significance. Source data are provided as a Source Data file.

Universal qPCR Master Mix (NEB, M3003L). Sequences of all primers used can be found in Suppl. Dataset 9.

## Immunoblotting, immunofluorescence, and immunohistochemistry

For Western blot assays, according to our prior publication[58–60], cultured cells were washed with ice-cold PBS and lysed in cell lysis buffer. Whole cell lysates were subjected to SDS–PAGE and protein expression was visualized using an enhanced chemiluminescence detection system (GE Healthcare, Little Chalfont, UK) as described before[55]. The following dilutions were used for each antibody: JUNB (Cell Signaling Technology, #3753; 1:1000), Histone 3 (Santa Cruz, sc-56616; 1:5000). For SA-β-GAL staining, the β-galactosidase Senescence Kit (Cell Signaling Technology, #9860) was used. Briefly, cells were fixed for 15 min followed by washing in PBS twice. Fixed cells were then incubated with β-galactosidase staining solution at 37 °C overnight in a dry incubator (no $CO_2$). The cells were then observed and counted for the SA-β-GAL positive cells. For immunofluorescence staining, cultured cells were fixed in 4% PFA for 15 min and blocked with 3% BSA within 1 h. Primary antibodies were applied to samples with indicated dilution below and the samples were kept at 4 °C overnight. For immunofluorescence staining[57,61], cultured cells or myofibers were fixed in 4% PFA for 15 min and permeabilized with 0.5% NP-40 for 10 min. Then cells were blocked in 3% BSA for 1 h followed by incubating with primary antibodies overnight at 4 °C and secondary antibodies for 1 h at RT. Finally, the cells were mounted with DAPI to stain the cell nucleus and images were captured by a Leica fluorescence microscope. Primary antibodies and dilutions were used as following: PAX7 (Developmental Studies Hybridoma Bank; 1:50), P16 (Abcam, ab211542, 1:200). For immunohistochemistry[48,57,61], in brief, slides were fixed with 4% PFA for 15 min at room temperature and permeabilized in ice-cold methanol for 6 min at −20 °C. Heat-mediated antigen retrieval with 0.01 M citric acid (pH 6.0) was performed for 5 min in a microwave. After 4% BSA (4% IgG-free BSA in PBS; Jackson, 001-000-162) blocking, the sections were further blocked with unconjugated AffiniPure Fab Fragment (1:100 in PBS; Jackson, 115-007-003) for 30 min. The biotin-conjugated anti-mouse IgG (1:500 in 4% BBBSA, Jackson, 115-065-205) and Cy3-Streptavidin (1:1250 in 4% BBBSA, Jackson, 016-160-084) were used as secondary antibodies. Primary antibodies and dilutions were used as follows: Laminin (Sigma-Aldrich L9393-100UL, 1:800), P16 (Abcam, ab270058, 1:200), P21 (Santa Cruz Biotechnology, sc-6246, 1:200) PAX7 (Developmental Studies Hybridoma Bank; 1:50) for staining of muscle cryosections. Images were slightly modified with ImageJ in which background was reduced using background subtraction and brightness and contrast were adjusted. H&E (Hematoxylin and eosin), was performed as previously described[55,61,62].

## Multiome snRNA-seq/ATAC-seq profiling in human muscle

snRNA-seq/ATAC-seq and scRNAseq were performed on 10x genomics platform. Briefly, mononucleated resident cells were isolated from human muscle as described in "Fluorescence-activated MuSC sorting and culturing" part with 7-AAD (Thermo Scientific, 00-6993-50) staining for viability selection. Red blood cells were eliminated by ACK buffers (150 M $NH_4Cl$, 100 mM $KHCO_3$, 10 mM EDTA-2Na) before sorting. After sorting, live cells were washed with 0.04% BSA in PBS twice and resuspended in the BSA solution. For snRNA-seq/ATAC-seq, nuclei were isolated from the suspended cells according to the manufacturer's instruction CG000366 • Rev D. Isolated nuclei were counted under a microscope and Typan blue was used to examine the number and integrity. The isolated nuclei were then resuspended at an appropriate concentration (5000−10000 nuclei/μl); Library construction was performed following the manufacturer's instructions for generation of Gel Bead-In Emulsions (GEMs) using the 10x Chromium system.

## Initial processing and quality control of snRNA-seq and snATAC-seq data

Raw sequencing reads of human skeletal muscle were aligned to the pre-built reference on GRCh38 and counted using Cell Ranger ARC (version 2.0.1) with the default parameters. High-quality nuclei were kept based on gene expression data (>1000, and <40,000 UMI, and mitochondrial percent <20) and chromatin accessibility data (>1000, and <100,000 ATAC read counts). Seurat (version 5.0.1) object of each sample was constructed from clean nuclei and merged using.

## Integration, clustering and identification of cell types

As the standard pipeline of Seurat and Signac (version 1.12.0) package recommended, we next performed pre-processing and dimensional reduction on both assays independently. First, the RNA count matrix of each sample was normalized using the SCTransform function with the mitochondrial percent variable regressed out. To match shared cell types across samples, features and anchors for downstream integration were selected with the *FindIntegrationAnchors* and *IntegrateData* functions, ensuring accurate comparative analysis. After data integration and scaling, principal component analysis (PCA) was conducted on a new integrated assay with the RunPCA function, and clustering and dimensionality reduction analysis was performed with the *FindNeighbors*, *FindClusters*, and *RunUMAP* functions. Cell types were identified and annotated according to the expression levels of the classic marker genes. The marker genes of each cell type were calculated using the *FindAllMarkers* function with the cutoff of LogFC >1 and adjusted *P*-values <0.05 using *t*-test.

Second, the ATAC counts of each sample were normalized by *RunTFIDF* function. LSI coordinates at the sample level were computed on the normalized ATAC matrix using *RunSVD*. To identify integration

anchors, different samples were projected into a shared space by *FindIntegrationAnchors* function. and the low-dimensional cell embeddings (the LSI coordinates) across the datasets were integrated using the *IntegrateEmbeddings* function. UMAP clustering was created using the integrated embeddings by *RunUMAP* function.

A weighted nearest neighbor (WNN) graph was constructed by *FindMultiModalNeighbors* from a list of two-dimensional reductions: PCA from RNA assay and integrated LSI from ATAC assay. WNN graph was used for following UMAP visualization and clustering.

### Transcriptional noise analysis

Transcriptional noise/heterogeneity analysis was conducted following previous work[63,64]. To account for differences originating from UMI counts and cell-type composition in different age groups, all cells were down-sampled to a specified number of UMIs, cell numbers were then down-sampled so that equal numbers of young and aged cells were used. A list of representative invariant genes was selected as the prior method suggested to calculate the Euclidean distance from each cell to the corresponding cell-type mean vector within each age group[63]. Additionally, the Euclidean distances were averaged for each donor from two aged groups to further remove technical confounding. The Euclidean distance was used to measure the transcriptional noise at both single-cell and cell-type levels.

### Aging sensibility analysis

The prioritization of cell types in the response to human muscle aging was calculated and named as Augur Score using *calculate_auc* function from Augur package (version 1.0.3) by inputting the genes-by-cells scRNA-seq matrix and a data frame containing cell type and aged group columns.

### Senescent cell identification

To define senescent cells, we developed a unified senescence scoring (USS) algorithm based on five established senescence gene databases (SM: SenMayo[30], CA: CellAge[29], GA: GenAge[28], and SE: Senescence Eigengene approach[65]). (Suppl. Dataset 2). All cells were first divided based on cell types. Within the same cell type, ss-GSVA score was calculated for each cell with GSVA package (version 1.42.0) using the four different gene sets. Note that we intentionally excluded known senescent markers (*P16*, *P15*, *P19*, *P21*, *P27*, and *PAI-1*) from all four gene lists, since these genes will serve as additional validation of senescence signatures in the detected cell subset. ss-GSVA score SM, CA, GA, and SE were each split into two halves by the median value and senescent cells were defined as those possessing ss-GSVA scores in the upper half level.

### Differential expression and Gene Ontology enrichment analysis

Differentially expressed genes (DEGs) were determined in senescent vs. non-senescent for each cell type. DEGs were analyzed by *FindMarkers* function in Seurat using Wilcoxon Rank Sum test, and were detected with the cutoff of LogFC >0.5 and adjusted *P*-value <0.05. Sn-DEG lists for each cell type are shown in Suppl. Dataset 3. Gene Ontology (GO) enrichment analysis for Sn-DEG sets was performed with the *enrichGO* function in the clusterProfiler (version 4.2.0) package. GO biological terms with Benjamini-Hochberg adjusted *P*-value (FDR) <0.05 were considered significantly enriched.

### Cell fate trajectory analysis

To infer the aging pace for selected cell types, Monocle (version 2.14.0) package was used for cell trajectory and pseudotime analysis[66]. Briefly, for each cell cluster from two age groups, the Sn-DEGs in the cell type were used as ordering genes for DDRTree analysis by *reduceDimension* function and pseudotime ordering by *orderCells* function. To identify genes with expression patterns positively or negatively linked to pseudotime scale, Spearman correlations between the pseudotime value and gene expression levels were calculated among cells clustered along the pseudotime trajectory. Genes with high correlation with pseudotime scale were visualized with smooth expression curves by *plot_pseudotime_heatmap* function in Monocle package.

For MuSCs, the cells falling into late 1 and late 2 branches were extracted, and DEGs between the two late fates were detected (late branch-DEGs). Furthermore, late branch-specific DEGs significantly linked to pseudotime at late branch stage were identified and visualized in a similar manner.

To further examine the senescence characteristics along the pseudotime trajectory, cell cycle activity signature was defined as ss-GSVA score for cell cycle gene list from REACTOME knowledgebase[39].

### Cell-cell communication analysis

Cell-cell interactions were inferred by CellChat (version 1.6.1) based on the expression of known ligand-receptor pairs in various cell types[67]. Cells from young and aged groups were applied to CellChat separately and merged into one CellChat object. Dysregulated signaling during aging is identified by i*dentifyOverExpressedGenes* and *identifyOverExpressedInteractions* functions. Age group-specific pathways and ligand-receptor pairs were also detected and visualized using functions wrapped in CellChat.

### DNA sequence motif enrichment and TF regulation analysis

ATAC peaks were called for each cell type using Signac and used in subsequent analyses retaining the cell type annotations[68]. To search and compute enriched motifs, the DNA sequence of each peak was scanned, and a motif object was created and added to the Seurat object. Per-cell accessibility scores for known motifs were calculated and stored as a new assay (chromvar) by RunChromVAR function wrapped in chromVAR package (version 1.16.0). The chromvar assay contained chromVAR motif accessibilities and facilitated the identification of regulators of senescent cell state. Putative TF regulators were defined as those with significantly higher accessibility scores in senescent compared to non-senescent cells by *wilcoxauc* function from presto package (version 1.0.0).

### Transcriptional regulatory network analysis

To infer putative peak-gene regulatory interactions from paired snATAC-seq and snRNA-seq data, the distal cis-regulatory elements significantly associated with genes were computed by FigR[38]. DORC (domains of regulatory chromatin) analysis was conducted to assess the accessibility of peaks within a fixed window (100 kb) centered around the transcription start site (TSS) of specific target genes and correlated with their expression levels. By combining the significance estimates of relative motif enrichment and RNA expression correlation for a given DORC, a signed regulation score (RS) was calculated, with the sign indicating whether the TF acts as an activator or repressor. TF-gene networks were then inferred to pinpoint candidate TF regulators. Specifically, JUNB DORC accessibility was calculated as the accessibility of JUNB motif-containing peaks in each SASP target gene, to show that JUNB accessibility can predict SASP gene expression along senescence trajectories. To visualize dynamics of DORC accessibility and gene expression of JUNB-target SASP genes along the pseudotime axis, we used the *genSmoothCurves* function to fit smooth spline curves for JUNB DORC accessibility and gene expression matrix dynamics along aging pseudotime on a gene-wise basis. Subsequently, these matrices were normalized to the 1-99 percentile values respectively, to the relative difference between DORC and RNA. Furthermore, we applied a loess smoothing function to the normalized DORC/RNA values in relation to the smoothed aging pseudotime, which was then overlaid and visually represented.

## scRNA-seq RNA-seq and data analysis

For single-cell RNAseq profiling in DMSO and MVC-treated mice, mononucleated cells were sorted as described in "Fluorescence-activated MuSC sorting and culturing" part with 7-AAD (Thermo Scientific, 00-6993-50) staining for viability selection. After sorting, cells were resuspended in the BSA solution at an appropriate concentration (800–1200 cells/µl). Suspended cells were counted under a microscope and Typan blue was used to examine the cell viability. Library construction was performed following the manufacturer's instructions for generation of Gel Bead-In Emulsions (GEMs) using the 10x Chromium system.

To analyze the above generated scRNA-seq data, cells with low gene number (fewer than 500) and high ratio of mitochondrial genes (more than 10%) were first removed. SCTransform normalization, clustering, and cell-cell interaction analysis were conducted in a similar manner as the snRNA-seq data analysis described above.

## Bulk RNA-seq and data analysis

For conducting RNA-seq (polyA + mRNA) in MuSCs from DMSO and MVC-treated mice, following our prior protocol[43,48] total RNAs were subjected to polyA selection (Ambion, 61006) followed by library preparation using NEBNext Ultra II RNA Library Preparation Kit (NEB, E7770S). Libraries were paired-end sequenced with read lengths of 150 bp on Illumina Nova-seq S4 instruments. The raw reads of RNA-seq were processed following the procedures described in our previous publication[56]. Briefly, the adapter and low-quality sequences were trimmed from 3' to 5' ends for each read, and the reads shorter than 36 bp were discarded. The clean reads were aligned to mouse (mm10) reference genome with STAR. Next, Cufflinks was used to quantify the gene expression. Genes with expression level change >1.5-fold and adjusted $p$-value <0.1 were identified as DEGs between two conditions. GO enrichment analysis was performed using R package clusterProfiler.

## CUT&RUN-seq and data analysis

CUT&RUN assay was conducted following our prior protocol[48] using 200,000 MuSCs cells with the CUT&RUN assay kit (Cell Signaling Technology, 86652). In brief, FISCs were harvested and washed by cell wash buffer, then bound to concanavalin A-coated magnetic beads. Digitonin Wash Buffer was used for permeabilization. After that, cells were incubated with 2 µg of JUNB antibody (Cell Signaling Technology #3753) or H3K27ac (Cell Signaling Technology #8173) overnight at 4 °C with shaking. Then, cell bead slurry was washed with Digitonin Wash Buffer and incubated with Protein A-MNase for 1 hr at 4 °C with shaking. After washing with Digitonin Wash Buffer, CaCl2 was added into the cell-bead slurry to initiate Protein A-MNase digestion, which was then incubated at 4 °C for half an hour. Then 2x Stop Buffer was added to the reaction to stop the digestion. CUT&RUN fragments were released by incubation for 30 min at 37 °C followed by centrifugation. After centrifugation, the supernatant was recovered, and DNA purification was performed by using Phenol/Chloroform. For DNA library construction, a NEBNext® Ultra™ II DNA Library Prep Kit for Illumina® (NEB, E7645S) was used according to the manufacturer's instructions. Bioanalyzer analysis and qPCR were used to measure the quality of DNA libraries including the DNA size and purity.

The obtained sequencing raw reads were first pre-processed by quality assessment, adapters trimming, and low-quality filtering, and then were aligned to the mouse reference genome (mm10) using Bowtie2, and only non-redundant reads were kept. JUNB binding sites (JUNB peaks) were identified with $p$-value cutoff as 0.001 by MACS2[69] and genome browser visualization files were generated by Homer[70]. To further investigate the JUNB binding sites, we used H3K27ac signal to indicate enhancer regions using our H3K27ac CUT&RUN-seq data in mouse MuSC from UCSD Human Reference Epigenome Mapping Project[71].

## Statistics and reproducibility

Data represent the average of at least three independent experiments, humans or mice + s.d. unless indicated. The statistical significance of experimental data was calculated by the Student's $t$-test (two-sided). *$p$ <0.05, **$p$ <0.01, ***$p$ <0.001, ****$p$ <0.0001 and $n.s.$: no significance ($p ≥ 0.05$). The statistical significance for the assays conducted with MuSCs from the same human or mouse with different treatments was calculated by the student's $t$-test (paired). *$p$ <0.05, **$p$ <0.01, ***$p$ <0.001, n.s. no significance ($p ≥ 0.05$). Specifically, a single zero-truncated negative binomial distribution was fit to the input data and each region was assigned a $P$-value based on the fitted distribution. Representative images of at least three independent experiments are shown in Figs. 2L–O; 5E; and Supplementary Figs. 4H; 5C, J, Q; 7F, J.

## Reporting summary

Further information on research design is available in the Nature Portfolio Reporting Summary linked to this article.

## Data availability

The human skeletal muscle single-nucleus multiome within this study have been deposited in the Gene Expression Omnibus database under the accessions GSE268953. The mouse single-cell RNA-seq, bulk RNA-seq, and CUT&RUN datasets have been deposited under the accessions GSE268407, GSE268952, and GSE268433, respectively. Source data are provided with this paper. All other data supporting the findings of this study are available from the corresponding author on reasonable request. Source data are provided with this paper.

## Code availability

The code used in this study is available at the GitHub repository https://github.com/Hannah-bioinfo/Scripts_Aging_SnC_MS/.

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

## Acknowledgements

This work was supported by Non-Communicable Chronic Disease-National Science and Technology Major Project of China to H. W. (project code: 2024ZD0530400); National Key R&D Program of China to H.W. (project code: 2022YFA0806003); The InnoHK initiative of the Innovation and Technology Commission of the Hong Kong Special Administrative Region Government to H.W. and T.X. (project code:INNOHK22SC01); Health and Medical Research Fund (HMRF) from Health Bureau of HK to H.W. (project codes: 10210906 and 08190626); Theme-based Research Scheme (TRS) from RGC to H.W. (project code:T13-602/21-N); General Research Fund (GRF) from Research Grants Council (RGC) of the HongKong Special Administrative Region, China to H.W. (project codes: 14108225, 14106521, 14105123, 14103522, and 14105823 to H.W.); the National Natural Science Foundation of China (NSFC) to H.W. (project codes: 82172436); Area of Excellence Scheme (AoE) from RGC to H.W. (project code: AoE/M-402/20). The Chinese University of Hong Kong (CUHK) Strategic Seed Funding for Collaborative Research Scheme (SSFCRS) to H.W.

## Author contributions

Yang Li performed most of the wet-lab experiments; Chuhan Li analyzed all the high-throughput sequencing data; Xingyuan Liu, Qin Zhou and Yulong Qiao performed and helped with animal experiments, human muscle section staining, single-cell RNA-seq and JunB CUT&RUN-seq; Ting Xie provided aged mice; Michael Tim-Yun Ong provided the human muscle specimens; Hao Sun supervised computational analyses; Huating Wang supervised experiments; Yang Li, Chuhan Li and Huating Wang conceived the project and wrote the manuscript, with inputs from all authors.

## Competing interests

The authors declare no competing interests.
