## [Transparent Peer Review file · Nature Communications]

Multiomics mapping and characterization of cellular senescence in aging human skeletal muscle uncovers a novel senotherapeutic for sarcopenia

Corresponding Author: Dr Huating Wang

Version 0:

Reviewer comments:

Reviewer #1

(Remarks to the Author)

In the manuscript titled "Multiomics mapping and characterization of cellular senescence in aging human skeletal muscle uncovers a novel senotherapeutic for sarcopenia", Li Y & Li C et al leveraged simultaneous snRNA-seq and ATAC-seq on the mononucleated cells in skeletal muscle from young and old human subjects and identified the senescence signatures in the different cell populations in skeletal muscle. The authors identified the CCL-family chemo-attractants to be highly induced in senescent cells. The authors then showed that targeting CCR5 using Maraviroc can lead to improved muscle performance and reduced SASP signaling in various cell types in mice. The authors further found that ATF3 targets were enriched in the ATAC-seq data and JUNB is the potential activator. The multiomics data generated by this study is comprehensive and will be useful for the muscle and aging community. The authors also demonstrated the possibility of Maraviroc and JUNB as the senotherapeutic strategy. The study is in general well designed and executed, and the conclusions are largely supported by the data. Overall, this is an exciting human study with a few minor weaknesses that can be improved in a revision.

Specific concerns for improvement:

1. The authors mentioned that the 10 human muscle hamstring biopsies were from 5 young men who underwent anterior cruciate ligament reconstruction and 5 aged men who underwent knee replacement surgery. While it is understandable that these are not perfect controlled treatment groups due to ethical regulation on human sample collection, it would be nice to justify the rationale and state the weakness of the sampling design. Information about the severity of the disease condition on these human subjects and the different degree of underlying inflammation/damage in the muscle in the two groups would be important for interpretation of the results. For example, is there information on the basal level of senescence/inflammation in these subjects? Are these effects taken into consideration in data interpretation?
2. Figure 2A: The authors separated the EC population into 3 clusters, the SMCs into 2 clusters and the FAPs into 2 clusters. Yet in Figure 2B & 2C, the author used only one cluster for EC, SMC and FAP. Based on Figure 2D, there are different transcriptional noises in the different EC, SMC, FAP clusters shown by the authors. How different are these EC/SMC/FAP clusters and do some of the clusters represent the senescent population?
3. Figure 4I, in the RT-qPCR result, can the authors explain the data analysis method? Why did the young group not have an error bar?
4. In the Maraviroc treatment study, how did the overall SA-b-Gal staining change after the treatment and what were the changes to p16, p21 etc in the MuSCs and other cells? This information would be beneficial to establish the senotherapeutic potential of Maraviroc.
5. In Figure 3A-D, the authors described two late fates which are both senescent but have different transcriptional profiles. When MVC was administered to the animals, does it alter the two late fates differentially?
6. Can the authors comment on the findings that the SASP targets (DCN, VCAN, CXCL2, CCL2, CXCL1) identified by

snRNA-seq in MuSCs do not overlap with the JUNB targets (IL1R1, TGFB3, TNFRSF10D, TNFRSF1A, GDF15 etc) identified by ATAC-seq? Does this result suggest both JUNB dependent and independent regulation of SASP associated genes?

Other minor concerns:

1. Typo in line "In the first high dose short term (HDHT) treatment regime...", "HDHT" should be "HDST".
2. What does "delay the degree of senescence" mean?

Reviewer #2

(Remarks to the Author)

In this manuscript, Li and colleagues explore cellular senescence in human (and) mouse skeletal muscle. This manuscript is quite extensive. They use multiomics approaches single nuclei RNAseq and ATACseq to identify and characterize senescent cells in the different mononucleated population present in the tissue and their associated secretory phenotype (SASP). With the RNAseq analysis they identify CCR5 which they target in mice using maraviroc and showed an increased muscle mass and strength compared to DMSO treated control.

They also analyzed their ATACseq data and identified ATF3 and JUNB as potential TF regulators in senescent cells. However, this part is not really linked with the first part of the manuscript.

Major comments:

It is well known and the authors mentioned in their introduction that senescence is very heterogeneous process but at time notably when they performed their validation experiments they used the global population of MuSCs isolated from aged donor without mentioning clearly in their analysis that not all the cells there are senescent. For example, in fig2f the analysis of the rnaseq data analysis for senescence signature showed the smallest increase in the musc cluster 12.9% vs 7.9 in young but all their further validation isolating MuSCs from aged animal and testing p16/p21 showed drastic increase in those marker and the author do not comment on this difference.

In fig5 after identification of CCR5 as potential target, they propose to test Maraviroc as a potential senotherapeutic drug. To test it potential they switch model and make use of mouse model. Since there is a change in species before any functional test I would have expected some RTqPCR or analysis of published RNAseq data set showing that similar observation can be also shown in mice : increase in CCR5 and eventually also the CCL3,4,5.

Finally to conclude that it is a senotherapeutic effect I would also have included a cohort of young animal following the same treatment and analysis of the tissues to show eventual response specificity in the aged cohort.

Finally ATACseq analysis identify JUNB as a potential regulators although the loss or gain experiments with mJunB did not appear to affect the levels of SA- β -GAL or marker genes such as p16, p19, p21 and p53 (Suppl. Fig. S7E-K), suggesting mJunB is not sufficient to trigger full senescence in mouse MuSCs. The further validation in fibroblast instead of myoblast are not as relevant, immortalized human myoblast could have been used.

This part is a bit disconnected from the rest as there is no directly link to the target identified with the first part of the study. Does ATF3 or JUNB regulate CCR5 or CCL3,4,5?

Finally the authors to obtain their single cell suspension before RNAseq and further validation experiments used traditional isolation method based on collagenase dissociation but multiple labs particularly in the muscle field have reported that this can lead to aberrant expression of certain early stress genes notably members of the AP1 complex.

(10.1016/j.celrep.2017.10.080 ;

10.1016/j.celrep.2017.10.037 ; 10.1038/nmeth.4437).The authors do not mention those studies it would be good to comment or showed that this did not apply in their setup.

Minor comments:

"Not only MuSC activities are largely impacted by the signals from the niche[15], MuSCs can also actively modulate the niche through its immune-secretory function[16, 17]. Moreover, burgeoning evidence suggested signs of senescence in skeletal muscle but conflicting observations are being reported[12, 18]. For example,earlier studies reported signs of senescence in MuSCs from aged mice (elevated expression of SA- β -GAL and p16, p21, and Igfbp5 mRNAs) but recent results[19, 20] claim no solid evidence of senescence in intact muscles. By contrast, senescence in FAPs increases with muscle aging and is shown to facilitate muscle regeneration[19]"

This part of the introduction was confusing, I understand it presents some conflicting studies but it needs some clarification. In fig 2d authors want demonstrate a higher transcriptional noise/heterogeneity in aged samples vs young one but the fig 2e plotting the different cell type ratio tends to show often higher variation in the young sample distribution particularly regarding

satellite cells. The authors dont comment on this.

Reviewer #3

(Remarks to the Author)

Version 1:

Reviewer comments:

Reviewer #1

(Remarks to the Author)

In the revision for the manuscript "multiomics mapping and characterization of cellular senescence in aging human skeletal muscle uncovers a novel senotherapeutic for sarcopenia" by Li Y & Li C, the authors have satisfactorily answered the questions raised by the reviewer. This reviewer do not have further questions for the authors.

Reviewer #2

(Remarks to the Author)

The authors responded comprehensively to the comments of all reviewers. They provided the requested extra analyses/experiments and adjusted part of the text according to comments. The manuscript is now ready for publication.

Reviewer #3

(Remarks to the Author)

Point-to-point letter

Reviewer #1

In the manuscript titled "Multiomics mapping and characterization of cellular senescence in aging human skeletal muscle uncovers a novel senotherapeutic for sarcopenia", Li Y & Li C et al leveraged simultaneous snRNA-seq and ATAC-seq on the mononucleated cells in skeletal muscle from young and old human subjects and identified the senescence signatures in the different cell populations in skeletal muscle. The authors identified the CCL-family chemo-attractants to be highly induced in senescent cells. The authors then showed that targeting CCR5 using Maraviroc can lead to improved muscle performance and reduced SASP signaling in various cell types in mice. The authors further found that ATF3 targets were enriched in the ATAC-seq data and JUNB is the potential activator. The multiomics data generated by this study is comprehensive and will be useful for the muscle and aging community. The authors also demonstrated the possibility of Maraviroc and JUNB as the senotherapeutic strategy. The study is in general well designed and executed, and the conclusions are largely supported by the data. Overall, this is an exciting human study with a few minor weaknesses that can be improved in a revision.

1.1 The authors mentioned that the 10 human muscle hamstring biopsies were from 5 young men who underwent anterior cruciate ligament reconstruction and 5 aged men who underwent knee replacement surgery. While it is understandable that these are not perfect controlled treatment groups due to ethical regulation on human sample collection, it would be nice to justify the rationale and state the weakness of the sampling design. Information about the severity of the disease condition on these human subjects and the different degree of underlying inflammation/damage in the muscle in the two groups would be important for interpretation of the results. For example, is there information on the basal level of senescence/inflammation in these subjects? Are these effects take into consideration in data interpretation?

A: Thank you for the comment. We apologize for the unclear description of the muscle source. Briefly, all subjects were confirmed free of preexisting severe myopathies and other related diseases. Structurally intact and histologically healthy hamstring muscles were harvested from anatomically equivalent regions across cohorts and processed under standardized protocol. This experimental design ensured that confounding variables derived from pathological states or procedural artifacts were minimized, thereby preserving tissue integrity for downstream analyses. As expected, evident inflammation was observed in aged but not young group (Fig. 2L). From analyzing snRNA-seq data, increased senescence was also detected in aged vs. young group (Fig. 2F-J). We have now revised the manuscript on page 6 and page 24 to include the needed information on human biopsies.

1.2 Figure 2A: The authors separated the EC population into 3 clusters, the SMCs into 2 clusters and the FAPs into 2 clusters. Yet in Figure 2B & 2C, the author used only one cluster for EC, SMC and FAP. Based on Figure 2D, there are different transcriptional noises in the different EC, SMC, FAP clusters shown by the authors. How different are these EC/SMC/FAP clusters and do some of the clusters represent the senescent population?

A: Thank you for your insightful comment. In Fig. 2A we identified subclusters of EC, SMC, and FAP using unsupervised clustering. These subclusters were closely positioned in the UMAP plot (Fig. 2A), indicating relatively small transcriptional differences. Thus, in Fig. 2B&2C these subclustered were merged to better show the differences among the main cell types. Nevertheless, we detected significant differences among

the subclusters when examining transcriptional noise (Fig. 2D). As suggested, we have now examined the senescence heterogeneity among the subclusters. Significantly different percentages of senescent cells were detected among the subclusters of each cell type (data not shown). For example, 23.1% of senescent cells were detected in FAP1 subcluster while only 2.5% in FAP2, indicating FAP1 could represent the senescent FAP population (see figures below). We have revised the text on page 7.

1.3 Figure 4I, in the RT-qPCR result, can the authors explain the data analysis method? Why did the young group not have an error bar?

A: Thanks for the comments. Because the muscle samples were not collected at the same time, we performed a two-sided paired Student t-test in Fig. 4I, which is usually used for the comparison of two groups [1, 2]. The values for young group were set to 1.0 therefore no error bars were necessary. We have now updated the description in the revised figure legend on page 40.

1.4 In the Maraviroc treatment study, how did the overall SA- β -Gal staining change after the treatment and what were the changes to p16, p21 etc. in the MuSCs and other cells? This information would be beneficial to establish the senotherapeutic potential of Maraviroc.

A: Thanks for the critical suggestion. Accordingly, we have now performed overall SA- β -Gal on muscle sections after the MVC treatment. We observed very limited SA- β -Gal positive areas in both DMSO/MVC treatment groups (data not shown). We think this was due to the technical difficulty of staining muscle sections with SA- β -Gal which was also reported by others [3]. However, the p21 staining on the muscle sections was successful and uncovered significantly declined p21 level after the MVC treatment (Suppl. Fig. 4F). Consistently, RT-qPCR performed in both muscles and MuSCs showed a significant decrease in p16 and p21 expression levels after the MVC treatment (Suppl. Fig. 4G-H). Additionally, our scRNA-seq data also revealed declined p21 expression in MuSCs, FAPs, ECs and SMCs (Fig. 5M-P). Altogether, these results strengthened the senotherapeutic potential of MVC treatment. The newly added results can be found on page 13.

1.5 In Figure 3A-D, the authors described two late fates which are both senescent but have different transcriptional profiles. When MVC was administered to the animals, does it alter the two late fates differentially?

A: Thank you for the insightful question. To address this, we have now performed pseudotime analysis of the single cell RNA-seq data from MuSCs after MVC treatment. The results revealed the cells progressed into one late fate after the MVC treatment but a middle branch emerged; a lower portion of senescent cells were detected in the late fate confirming the reduced senescence by the MVC treatment (Suppl. Fig. 4J). The

newly added results can be found on page 13 of the revised manuscript.

1.6 Can the authors comment on the findings that the SASP targets (DCN, VCAN, CXCL2, CCL2, CXCL1) identified by snRNA-seq in MuSCs do not overlap with the JUNB targets (IL1R1, TGFB3, TNFRSF10D, TNFRSF1A, GDF15 etc) identified by ATAC-seq? Does this result suggest both JUNB dependent and independent regulation of SASP associated genes?

A: Thank you for your insightful question. In Fig. 4G, we only listed the top-ranked SASP in MuSCs, similarly only the top JUNB SASP targets were listed in Fig. 7A, therefore not much overlapping. The full lists can be found in Suppl. Table 4 and 7. In fact, the JUNB SASP targets were obtained by overlapping the predicted JUNB targets with the SASP list therefore all JUNB SASP targets should be found in the full SASP list. Nevertheless, we agree that both JUNB dependent and independent regulation of SASP genes exists as JUNB is only one of the identified potential SASP regulators (Fig. 6H-I, 7A). We have now revised on page 16 to make this clear.

1.7 Typo in line "In the first high dose short term (HDHT) treatment regime...", "HDHT" should be "HDST".

A: Thanks for your comment. We have corrected the typo on page 12.

1.8 What does "delay the degree of senescence" mean?

A: Thanks for your comment. We apologize for the unprecise description. It was suggested by Reviewer 2 in his/her comment 2.3 that study in Fig. 8 is a bit disconnected from the rest of the study (see below), so we have now removed this part from the manuscript and saved it for the next chapter of the study. We hope this is acceptable to the reviewer.

Reviewer #2

In this manuscript, Li and colleagues explore cellular senescence in human (and) mouse skeletal muscle. This manuscript is quite extensive. They use multiomics approaches single nuclei RNAseq and ATACseq to identify and characterize senescent cells in the different mononucleated population present in the tissue and their associated secretory phenotype (SASP). With the RNAseq analysis they identify CCR5 which they target in mice using maraviroc and showed an increased muscle mass and strength compared to DMSO treated control. They also analyzed their ATACseq data and identified ATF3 and JUNB as potential TF regulators in senescent cells. However, this part is not really linked with the first part of the manuscript.

2.1 It is well known and the authors mentioned in their introduction that senescence is very heterogeneous process but at time notably when their performed their validation experiments they used the global population of MuSCs isolated from aged donor without mentioning clearly in their analysis that not all the cells there are senescent. For example, in fig2f the analysis of the rnaseq data analysis for senescence signature showed the smallest increase in the musc cluster 12.9% vs 7.9 in young but all their further validation isolating MuSCs from aged animal and testing p16/p21 showed drastic increase in those markers and the author do not comment on this difference.

A: Thanks for your critical comment. We agree that using the senescent cells instead of the total population of MuSCs is a better choice for the validation experiments. We in fact made substantial efforts to isolate senescent MuSCs via SPiDER- β Gal FACS [3], which was however not successful. We have clarified this on page 8 that total MuSCs were used in the experiments. The discrepancy in the percentage of senescent cells detected by different methods may arise from the distinct sensitivities of these methods. However, all the results demonstrated an increasing trend.

2.2 In fig5 after identification of CCR5 as potential target, they propose to test Maraviroc as a potential senotherapeutic drug. To test it potential they switch model and make use of mouse model. Since there is a change in species before any functional test I would have expected some RTqPCR or analysis of published RNAseq data set showing that similar observation can be also shown in mice: increase in CCR5 and eventually also the CCl3,4,5. Finally to conclude that it is a senotherapeutic effect I would also have included a cohort of young animal following the same treatment and analysis of the tissues to show eventual response specificity in the aged cohort.

A: Thanks for the great comment. In the original manuscript, we have mentioned that “their induction was also found in aged mouse muscles by analyzing publicly available single-cell RNA-seq data” but did not show the data. We have now shown the upregulation of Ccr5 and Ccl5 in aged muscles and MuSCs (Suppl. Fig. 4C and Suppl. Fig. 4D, page 12). We have also performed RT-PCR to confirm the induction of Ccr5, Ccl4, and Ccl5 (Suppl. Fig. 4E, page 12). These data support the use of a mouse model to test the senotherapeutic effect of MVC.

We agree that including a cohort of young mice following the same treatment and analysis will show the specificity in the aged cohort. As suggested, we have now performed the high dose short duration MVC treatment in the young mice and found the treatment did not lead to significant alterations of muscle quality and function (Suppl. Fig. 5O-T, page 14), suggesting the treatment response is specific to the aged cohort.

2.3 Finally ATACseq analysis identify JUNB as a potential regulators although the loss or gain experiments with mJunB did not appear to affect the levels of SA- β -GAL or marker genes such as p16, p19, p21 and p53 (Suppl. Fig. S7E-K), suggesting mJunB is not sufficient to trigger full senescence in mouse MuSCs. The further validation in fibroblast instead of myoblast are not as relevant, immortalized human myobalst could have been used. This part is a bit disconnected from the rest as there is no directly link to the target identified with the first part of the study. Does ATF3 or JUNB regulate CCR5 or CCL3,4,5?

A: Thanks for your critical comment. We apologize for the unclear writing. Indeed, our data in Fig. 7 identified JunB as a SASP inducer but not a trigger of full senescence in mouse MuSCs. This is understandable considering cellular senescence is a complex multi-staged process orchestrated by complicated factors/pathways. The rationale for switching to human fibroblasts to test JunB regulation of SASP induction lies in the fact that we identified JunB as a potential SASP regulator not only in MuSCs but also in other cell types in aging muscle, leading us to speculate it may be a general regulator of SASP induction in many cells. This was further substantiated by recent publications showing AP-1 family as an inducer of SASP production in mouse and human fibroblasts [4-6]. However, we now agree with the reviewer that this part is a bit disconnected from the rest of the study, so we have decided to remove Fig. 8 from the manuscript. The general role of AP-1 in regulating senescence in other cells will be investigated in the next chapter of the study. We hope this is acceptable to the reviewer.

Interestingly, ATF3 or JUNB does not directly regulate CCR5 or CCL3, 4, 5. CCLs were identified as SASP targets in the first part of the study while JUNB was identified as the upstream SASP regulator in the second part. The two parts represent two directions of the study, together making the study multi-faceted that can benefit a wider range of readers. We have now revised the description on page 14 for easier understanding and hope this is acceptable to the reviewer.

2.4 Finally the authors to obtain their single cell suspension before RNAseq and further validation experiments used traditional isolation method based on collagenase dissociation but multiple labs particularly in the muscle field have reported that this can lead to aberrant expression of certain early stress genes notably members of the AP1 complex. (10.1016/j.celrep.2017.10.080 ; 10.1016/j.celrep.2017.10.037 ; 10.1038/nmeth.4437).The authors do not mention those studies it would be good to comment or showed that this did not apply in their setup.

A: Thanks for the insightful comment. We were aware that traditional isolation method based on collagenase dissociation can lead to aberrant expression of certain early stress genes notably members of the AP1 complex (we were one of the labs reporting this in Zhang S. 2023[7]). In fact, the single cell isolation can also induce AP1 expression. However, we reason that the same isolation method was used in both young and aged muscles in the validation experiments, therefore this should not bias our results leading to the conclusion that AP-1 functions as key regulators of SASP/senescence in aged MuSCs. We have now revised the text on page 22 to clarify the above point.

2.5 "Not only MuSC activities are largely impacted by the signals from the niche[15], MuSCs can also actively modulate the niche through its immune-secretory function[16, 17]. Moreover, burgeoning evidence suggested signs of senescence in skeletal muscle but conflicting observations are being reported[12, 18]. For example, earlier studies reported signs of senescence in MuSCs from aged mice (elevated expression of SA-β-GAL and p16, p21, and Igfbp5 mRNAs) but recent results[19, 20] claim no solid evidence of senescence in intact muscles. By contrast, senescence in FAPs increases with muscle aging and is shown to facilitate muscle regeneration[19]"

This part of the introduction was confusing, I understand it presents some conflicting studies but it needs some clarification.

A: Thanks for your suggestion. We apologize for the confusing writing. We have now revised text on page 4 to make it clear to read.

2.6 In fig 2d authors want demonstrate a higher transcriptional noise/heterogeneity in aged samples vs young one but the fig 2e plotting the different cell type ratio tends to show often higher variation in the young sample distribution particularly regarding satellite cells. The authors dont comment on this.

A: Thank you for your insightful comment. We apologize for the confusion. The two figures were in fact not related to each other. Fig. 2D illustrated the transcriptional noise in young (Y) versus aged (A) groups, measured by transcriptional differences with age at single-gene level. Fig. 2E showed the proportions in each cell type, revealing that the ratios of MuSC, EC, and SMC decrease with age while FAPs increase, measured by the cell number changes. The higher variation in the young sample distribution regarding MuSCs could be from the inherent differences among the 4 young samples used in the study. To avoid misunderstanding, we have now swapped the position of Fig. 2C with Fig. 2E (page 7).

Reference

1. Li, Y., et al., *Skeletal muscle stem cells modulate niche function in Duchenne muscular dystrophy mouse through YY1-CCL5 axis*. Nature Communications, 2025. **16**(1): p. 1324.
2. Zhao, Y., et al., *Multiscale 3D genome reorganization during skeletal muscle stem cell lineage progression and aging*. Science Advances, 2023. **9**(7): p. eabo1360.
3. Moiseeva, V., et al., *Senescence atlas reveals an aged-like inflamed niche that blunts muscle regeneration*. Nature, 2023. **613**(7942): p. 169-178.
4. Dasgupta, N., et al., *The role of the dynamic epigenetic landscape in senescence: orchestrating SASP expression*. 2024. **10**(1): p. 48.
5. Martínez-Zamudio, R.I., et al., *AP-1 imprints a reversible transcriptional programme of senescent cells*. 2020. **22**(7): p. 842-855.
6. Wang, Y., et al., *Unveiling E2F4, TEAD1 and AP-1 as regulatory transcription factors of the replicative senescence program by multi-omics analysis*. 2022. **13**(10): p. 742-759.
7. Zhang, S., et al., *ATF3 induction prevents precocious activation of skeletal muscle stem cell by regulating H2B expression*. 2023. **14**(1): p. 4978.